# Gene expression and functional deficits underlie TREM2-knockout microglia responses in human models of Alzheimer's disease

Amanda McQuade [1,2,3], You Jung Kang[4,5,6,7], Jonathan Hasselmann [1,2,3], Amit Jairaman[8], Alexandra Sotelo[2], Morgan Coburn[1,2,3], Sepideh Kiani Shabestari[1,2], Jean Paul Chadarevian[1,2,3], Gianna Fote[2,3], Christina H. Tu[2,3], Emma Danhash[2,3], Jorge Silva[2], Eric Martinez[2], Carl Cotman[3], G. Aleph Prieto[3,9], Leslie M. Thompson[1,2,3,5], Joan S. Steffan[2,3,10], Ian Smith [1], Hayk Davtyan [2,3], Michael Cahalan[8], Hansang Cho [4,5,6,7,11] & Mathew Blurton-Jones [1,2,3✉]

The discovery of TREM2 as a myeloid-specific Alzheimer's disease (AD) risk gene has accelerated research into the role of microglia in AD. While TREM2 mouse models have provided critical insight, the normal and disease-associated functions of TREM2 in human microglia remain unclear. To examine this question, we profile microglia differentiated from isogenic, CRISPR-modified TREM2-knockout induced pluripotent stem cell (iPSC) lines. By combining transcriptomic and functional analyses with a chimeric AD mouse model, we find that TREM2 deletion reduces microglial survival, impairs phagocytosis of key substrates including APOE, and inhibits SDF-1α/CXCR4-mediated chemotaxis, culminating in an impaired response to beta-amyloid plaques in vivo. Single-cell sequencing of xeno-transplanted human microglia further highlights a loss of disease-associated microglial (DAM) responses in human TREM2 knockout microglia that we validate by flow cytometry and immunohistochemistry. Taken together, these studies reveal both conserved and novel aspects of human TREM2 biology that likely play critical roles in the development and progression of AD.

[1] Department of Neurobiology & Behavior, University of California Irvine, Irvine, CA 92697, USA. [2] Sue and Bill Gross Stem Cell Research Center, University of California Irvine, Irvine, CA 92697, USA. [3] Institute for Memory Impairments and Neurological Disorders, University of California Irvine, Irvine, CA 92697, USA. [4] Department of Mechanical Engineering and Engineering Science, University of North Carolina Charlotte, Charlotte, NC 28223, USA. [5] Department of Biological Sciences, University of North Carolina Charlotte, Charlotte, NC 28223, USA. [6] Nanoscale Science Program, University of North Carolina Charlotte, Charlotte, NC 28223, USA. [7] Center for Biomedical Engineering and Science, University of North Carolina Charlotte, Charlotte, NC 28223, USA. [8] Department of Physiology and Biophysics, University of California Irvine, Irvine, CA 92697, USA. [9] Institute of Neurobiology, National Autonomous University of Mexico, Queretaro, Mexico. [10] Department of Psychology and Human Behavior, University of California Irvine, Irvine, CA 92697, USA. [11] Department of Biophysics, Sungkyunkwan University, Suwon 16419, Korea. ✉email: mblurton@uci.edu

Microglia and neuroinflammation are strongly implicated in the genetics and neuropathology of late-onset Alzheimer's disease (AD)[1,2]. As the primary immune cell of the brain, microglia perform key macrophage-functions, such as phagocytosis of dead cell debris and protein aggregates, cytokine/chemokine signaling, and immune surveillance and response (reviewed in McQuade et al.[3]). Yet, microglia also exhibit important neuroprotective functions that include providing trophic support for neurons, promoting oligodendrocyte differentiation, and modulating synaptic pruning and plasticity[3]. With the recent identification of several AD-risk loci near immune genes, combined with the development of pluripotent stem cell-derived microglia and chimeric mouse models, it is increasingly possible to examine the mechanisms that underlie the influence of human microglia on AD-risk.

Among these microglial-specific AD risk loci, variants in the triggering receptor expressed on myeloid cells 2 (TREM2) confer the largest effect on disease risk, causing a similar increase in risk as one APOE ε4 allele[4,5]. Within the central nervous system, TREM2 is predominantly expressed by microglia, raising fundamental questions about its role in driving microglia-specific functions associated with neurodegenerative diseases. TREM2 is part of the immunoglobulin superfamily and acts on the plasma membrane as a key member of the microglia sensome, mediating responses to phospholipids, APOE, and several other potential stimuli[6–9]. To transduce intracellular responses, TREM2 requires an adaptor protein, Tyro Protein Kinase Binding Protein (TYROBP/DAP12), which activates downstream signaling cascades including SYK, ERK, PLCG2, and NFAT.

Although complete loss-of-function mutations in TREM2 are most strongly linked to Nasu-Hakola disease and frontotemporal dementia, AD-associated TREM2 mutations such as R47H and R62H occur within the ligand-binding domain and are thought to confer a partial loss of function[10–13]. Some reports suggest that two loss-of-function TREM2 mutations, Q33X and W191X, may also influence AD risk[14,15], although this association remains unclear[16,17]. Nevertheless, studies of TREM2 deletion have greatly enriched our overall understanding of the normal function of TREM2 and have revealed important differences between murine and human cells in late versus early disease time-points[18–20].

For example, variable results have been reported regarding the impact of TREM2 deletion on tau pathology and neurodegeneration[18,21,22]. While the response of TREM2 knockout microglia to beta-amyloid is more consistent, in that murine microglia reliably show decreased clustering around plaques, the signals that underlie differential response remain unknown (reviewed in Hammond et al.[23]). Murine TREM2 has been shown to influence the accumulation of beta-amyloid in transgenic mice, although the magnitude and direction of this effect is variable and appears to be highly dependent on the stage of disease[18,19,22,24]. TREM2 has also been shown to influence microglial survival, although whether this function plays a role in human AD remains unknown[11,25]. To further understand these phenotypes and their impact on human disease, studies that examine the function of TREM2 in human microglia are critically needed.

In this study, we generated three isogenic TREM2 knockout induced pluripotent stem cell (iPSC) lines, differentiated these cells into microglia, and examined the transcriptional and functional effects of TREM2 deletion in human microglia. We have uncovered increased susceptibility to M-CSF-dependent survival in TREM2 KO microglia, decreased levels of phagocytosis of disease-relevant substrates, interactions between TREM2 and an allelic series of APOE, and CXCR4-dependent migration deficits. To further understand the impact of TREM2 deletion on human microglial function in vivo, we have combined this isogenic approach with a chimeric mouse model of AD[26], performing single-cell sequencing, flow cytometry, and neuropathological analysis to examine the interactions between microglia and beta-amyloid pathology in vivo, yielding a further understanding of the effect of TREM2 on human disease-associated microglia (DAM). Taken together, these studies provide insight into the normal and disease-associated functions of TREM2 in human microglia.

## Results

**Differential transcriptomic effects of TREM2 stimulation versus deletion in human iPSC-microglia.** Because preliminary data on TREM2 function suggests that AD-associated TREM2 mutations result in a partial loss of function[19], we sought to mimic this effect by generating three isogenic sets of CRISPR-modified TREM2 knockout (KO) iPSC lines. Following microglial differentiation[27], we verified the loss of TREM2 expression at the protein level in all knockout lines by western blot and homogeneous time resolved fluorescence (HTRF) analysis (Fig. 1a). HTRF analysis of conditioned culture medium further demonstrated secretion of soluble TREM2 exclusively in WT and not in TREM2 knockout cell lines (Fig. 1b). This strongly suggests that WT iPS-microglia exhibit normal trafficking and localization of TREM2, and further confirms lack of TREM2 expression in KO lines.

Next, we performed RNA sequencing to determine the genes and pathways that are disrupted in TREM2 KO human microglia. Two independent sets of TREM2 isogenic microglia generated from two distinct patients were sequenced to account for any cell line-dependent effects (Fig. 1c–e). Although some differences were noted between the two isogenic sets, many transcripts exhibited equivalent changes in both KO lines. Using a cutoff of ±1 log fold change and an FDR < 0.05, this dataset identified 390 differentially expressed genes (DEG) which were enriched for several important pathways including 'regulation of calcium-mediated signaling' and 'ERK1 and ERK2 cascades, and cell migration' (GO Biological Process 2018 adjusted $p < 0.05$; Fig. 1c–e and Supplemental Data 1). As expected, the most significantly downregulated gene in knockout microglia was TREM2 itself, followed closely by glycoprotein NMB (GPNMB) which was recently shown to be upregulated in plaque-associate microglia[28] (Fig. 1d and Supplemental Data 1). Interestingly, loss of GPNMB expression causes primary cutaneous amyloidosis[29], suggesting that GPNMB may play an important role in the clearance of amyloidogenic proteins. In line with recent AD mouse studies, TREM2 deletion also led to a significant reduction of APOE and APOC1 gene expression[12,30].

As some of the key downstream consequences of TREM2 expression in microglia may only manifest when TREM2 signaling is activated, we next sought to identify an optimal method to drive TREM2 signaling in wild type cells using TREM2 KO microglia as a negative control to further validate the specificity of this approach. A variety of ligands have been proposed to activate TREM2 downstream signaling, including APOE, beta-amyloid, phosphatidyl serine, and several other lipids[6–9], however, each of these ligands produces pleotropic effects, binding to and signaling through multiple receptors and pathways. Indeed, when TREM2 WT and KO cells are stimulated with dead iPSC-derived neuron fragments, both genotypes responded similarly (Supplemental Fig. 1 and Supplemental Data 2), suggesting that TREM2-specific responses to neuronal debris may be masked by signaling mediated via other microglia receptors, such as the TAM receptor tyrosine kinase family[31]. Because we found that TREM2 expression does not substantively alter the microglial transcriptional response to dead neurons, we

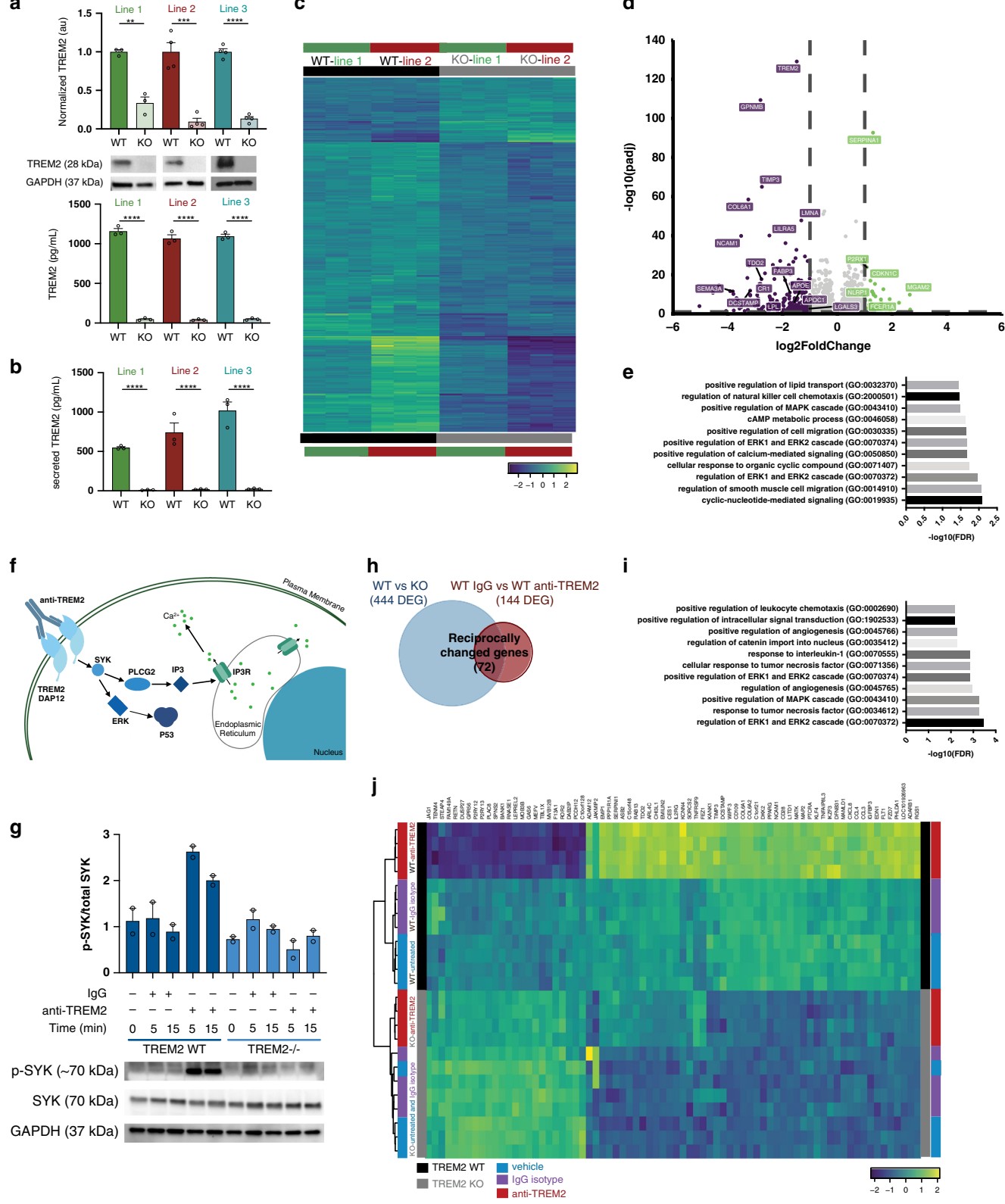

concluded that a more specific method of TREM2 activation would be needed to further interrogate TREM2 function.

Therefore, we opted to stimulate TREM2 signaling by using a commercially available polyclonal TREM2 antibody (R&D; AF1828). Similar protocols have proved useful for the activation of various immune receptors (e.g., antibody-based crosslinking of CD3 for T cell receptor activation), and this particular antibody

has been successfully used to quantify TREM2/DAP12 signaling with a split-luciferase reporter system[32]. Given the growing interest in development of TREM2 activating antibodies as a potential therapy for AD, this approach could also provide data that would further inform translational studies[33]. Since microglia express Fc receptors, an equivalent concentration of preimmune IgG was used as an additional isotype control. Using this

**Fig. 1 TREM2 knockout and stimulation elicit transcriptional changes in iPS-derived microglia. a** Confirmation of isogenic TREM2 knockout lines by western blot ($n = 3$ or 4 lanes for each of 3 lines; $t$-test **$p < 0.01$, Line 1 $p = 0.0024$, Line 2 $p = 0.0098$, Line 3 $p = 0.0094$), homogeneous time resolved fluorescence (HTRF; $n = 3$ independent wells for each of 3 lines; $t$-test ****$p < 0.0001$), and **b** sTREM2 secretion by HTRF; $n = 3$ independent wells for each of 3 $t$-test ****$p < 0.0001$). Different colors represent individual patient lines. Lighter shades represent KO lines. Data are represented as mean values ± SEM. **c** Heatmap of DEG for TREM2 WT versus KO ($n = 2$ independent lines; $n = 3$/line). Scale represents median-centered VST transformed counts from DESeq2. **d** Volcano plot showing DEG for TREM2 WT versus KO in 2 independent lines (green—increased expression in KO; purple—decreased expression in KO). **e** Geneset enrichment analysis of WT vs KO (EnrichR; adj $p < 0.05$ determined by EnrichR algorithm). Genes in these families were mainly increased in WT cells. **f** Schematic of TREM2 antibody stimulation paradigm. **g** Western blot showing phosphorylation of SYK in WT (dark blue) and KO (light blue) microglia within 5–15 min of exposure to the polyclonal TREM2 antibody (AF1828) or control IgG. (20 µg/mL[32] $n = 2$ independent samples). Data are represented as mean. **h** Venn diagram of differentially expressed genes in WT vs KO microglia (444 DEGs) compared to WT microglia 24 h after treatment with IgG versus anti-TREM2 antibody (144 DEGs). Venn diagram reveals 72 reciprocally changed DEGs. **i** Geneset enrichment analysis of reciprocally changed genes (EnrichR; adj $p < 0.05$ determined by EnrichR algorithm). **j** Heatmap of the resulting 72 reciprocally changed genes. Scale represents median-centered counts (TPM).

approach, we confirmed that AF1828 exposure produces rapid phosphorylation of spleen tyrosine kinase (SYK), a known signaling molecule downstream of TREM2/DAP12 activation (Fig. 1f, g). In contrast, treatment of WT cells with control IgG or treatment of TREM2 knockout microglia with AF1828 produced no phosphorylation of SYK (Fig. 1g).

Next, we coupled this stimulation approach with RNA sequencing to identify genes for which expression was altered both after knockout and, in the opposing direction, 24 h after antibody stimulation of WT microglia (Fig. 1h–j, Supplemental Data 3, log(FC) ≥ 0.5 and FDR < 0.001). These reciprocally changed genes are indicative of transcripts that, for example, decrease expression in the absence of TREM2 and increase expression in WT cells stimulated with anti-TREM2 antibody, but are not altered with control IgG or vehicle (DPBS) treatments. Geneset enrichment analysis on this highly specific reciprocally changed gene set showed similarity to enrichment seen with just knockout of TREM2 (Fig. 1e, j). The 72 reciprocally changed genes also highlighted key differences in chemotaxis (positive regulation of leukocyte chemotaxis), specific immune response families (cellular response to tumor necrosis factor), and cell survival pathways (positive regulation of ERK1 and ERK2 cascade) (Fig. 1i).

To functionally validate these findings, we next examined whether TREM2 expression and signaling alters microglial survival, phagocytosis, and chemotaxis as predicted by this geneset enrichment analysis.

**TREM2 knockout microglia are hypersensitive to stress-induced cell death**. From our previous geneset enrichment analysis, we found that MAPK and ERK pathways are strongly enriched (Fig. 1e, i). These pathways are involved in regulating cell survival and apoptosis. Additionally, previous data from murine microglia has indicated that TREM2 deficiency leads to increased cell death, a process that was hypothesized to rely on macrophage colony-stimulating factor (M-CSF)[34]. It has been suggested that TREM2 works cooperatively with CSF1R signaling which is necessary for microglial survival[35–37]. Therefore, we induced cell death in human iPS-microglia by cytokine starvation for 3 days and tested the importance of three key cytokines (M-CSF, IL-34, and TGF-β1) in microglial apoptosis (Fig. 2a).

Microglial apoptosis was visualized using a fluorogenic caspase 3/7 detector and time-lapse imaging. At baseline, we found TREM2 knockout microglia already showed increased caspase signal compared to their WT isogenic pairs which was not surprising given that MAPK and ERK pathways were enriched even when comparing WT and KO lines without stimulation (Figs. 1e, 2b).

Time-lapse imaging of WT microglia in complete medium (no cytokine starvation) revealed minimal caspase activation as

expected (Fig. 2a, c; dark blue). However, in TREM2 KO lines, increased apoptosis was quantified even in complete medium suggesting they may have a higher dependency on cytokines than WT lines (Fig. 2a, c; light blue). As expected, cytokine starvation (removal of M-CSF, IL-34, and TGF-β1) induced microglial death in both WT and KO lines. However, TREM2 KO lines still demonstrated elevated caspase 3/7 signal compared to WT lines (Fig. 2a, c; red), highlighting that TREM2 KO microglia exhibit increased sensitivity to stress conditions and respond through apoptosis.

In order to determine whether this apoptotic response occurs specifically through M-CSF/CSF1R signaling, iPS-microglia were grown in medium lacking *only* the CSF1R ligands (IL-34 and M-CSF). Indeed, this alone was sufficient to induce the same levels of apoptosis seen with full cytokine starvation (Fig. 2a, c; orange). In contrast, removal of TGF-β1 alone, an important regulator of microglial homeostasis, did *not* alter caspase activation (Fig. 2a, c; gray). Therefore, we conclude that human TREM2 modulates CSF1R signaling leading to higher levels of cell death in TREM2 knockout lines.

**TREM2 is necessary for phagocytosis of APOE by human microglia**. Apolipoprotein E (APOE), the largest genetic risk factor for AD, has been proposed as an important TREM2 ligand[5,12,38]. However, it remains unclear whether APOE-mediated disease risk is specifically related to its interactions with TREM2. Additionally, our sequencing data highlighted differences in lipid transport (Fig. 1e), prompting us to further examine the potential interactions between TREM2 and APOE. Therefore, we exposed TREM2 isogenic lines to an allelic series of recombinant, lipidated APOE (Fig. 3a and Supplementary Fig. 2).

Interestingly, we found that microglial phagocytosis of APOE in WT cells is dependent on APOE genotype, with APOE4 being internalized at a significantly higher rate than APOE3 which is taken up at higher levels than the AD protective allele; APOE2 (Fig. 3a). Importantly, these differences in phagocytosis were based on the APOE genotype of the exogenously added protein, not the APOE genotype of the microglia (all APOE 3/3). Further investigation is needed to determine if this recognition is crucial to AD development, but this data highlights that APOE genotypes are differentially recognized by microglia.

Next, we investigated the importance of TREM2 expression on phagocytosis of APOE. When TREM2 expression is lost, we detected no significant uptake of APOE fluorescence above the vehicle control (Fig. 3a). This stark difference was surprising given that TREM2 knockout microglia still express other canonical APOE receptors. Indeed, our RNA sequencing revealed no changes in expression of canonical APOE receptors between lines, including LDL2, LRP1, and HSPG2 (adj $p$-values 0.85, 0.98, 0.15, respectively) (Supplementary Data 1). Thus, even in the

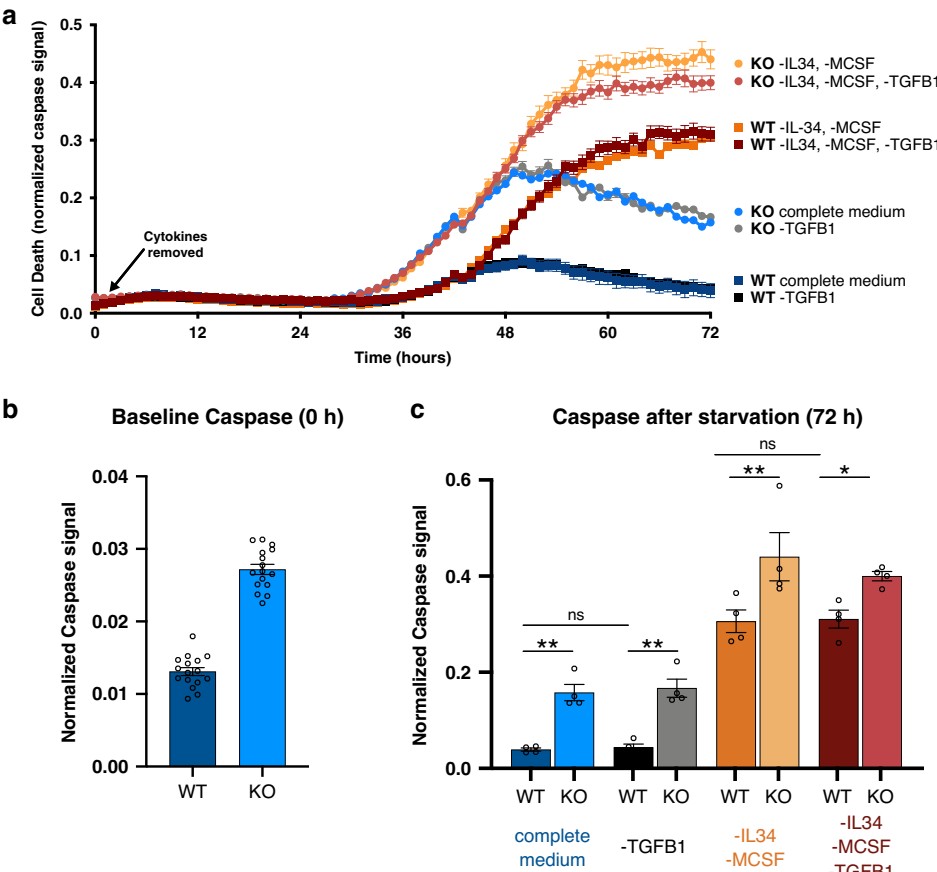

**Fig. 2 TREM2 knockout microglia exhibit increased caspase activation at baseline and after cytokine starvation. a** Caspase 3/7 levels imaged over 3 days in culture with complete medium (blue), no TGFB1 (gray), no IL-34, no MCSF (orange)or no IL-34, no MCSF, no TGFB1 (red). Images captured on Incucyte S3 live-cell imager. Darker shades represent TREM2 WT. Data are represented as mean values ± SEM. **b** Quantification of caspase 3/7 after 0 h in culture. (*t*-test ****p < 0.0001). Data are represented as mean values ± SEM. **c** Quantification of caspase 3/7 after 72 h in culture. (ANOVA, Tukey post hoc test. WT vs KO complete medium: *p* = 0.0052, WT vs KO -TGFB1: *p* = 0.0035, WT vs KO -IL34/MCSF: *p* = 0.0014, WT vs KO -IL34/MCSF/TGFB1: *p* = 0.045, ns: *p* > 0.9999). For all panels, *n* = 4 images in four independent wells. Data are represented as mean values ± SEM. Experiment was reproduced with two independent lines.

presence of other APOE receptors, TREM2 knockout microglia do not phagocytose detectable lipidated APOE protein suggesting that TREM2 is necessary for phagocytosis of APOE by human microglia.

To gain further mechanistic understanding on the role of TREM2 signaling in APOE phagocytosis, we also pre-treated iPS-microglia with a SYK inhibitor (see signaling schematic Fig. 1f), R406, to block TREM2 signal transduction. For these experiments, APOE3 was utilized as it represents the most common variant of APOE. Pre-treatment of microglia with R406 was able to partially, but significantly, block APOE phagocytosis in WT cells suggesting this phagocytosis does occur through TREM2/DAP12 signaling (Fig. 3b, right).

**TREM2 is critical for phagocytosis of fibrillar beta-amyloid and human synaptosomes but not Zymosan A.** As TREM2 has been proposed to act as a receptor for many ligands which are engulfed by microglia in vivo, we tested whether exposure of TREM2 isogenic lines to several other substrates would reveal corresponding functional differences. First, we exposed TREM2 isogenic lines to fibrillar beta-amyloid. As expected, phagocytosis of beta-amyloid was significantly decreased in KO lines (Fig. 1c). We further confirmed the requirement of TREM2/DAP12 downstream signaling by blocking SYK with R406 which proved

sufficient to block phagocytosis in WT lines. Importantly, blocking SYK in TREM2 KO lines had no effect (Fig. 1c).

To address previous studies that suggest TREM2 may play a role in synaptic pruning[39], we next exposed microglia to pHrodo-labeled human synaptosome fractions isolated from AD brain tissue[40]. Interestingly, TREM2 knockout microglia exhibited impaired phagocytosis of synaptosomes, suggesting that TREM2 may play an important role in synaptic pruning (Fig. 3d). Again, blocking downstream TREM2 signaling through SYK with R406 was able to partially block phagocytosis of synaptosomes in WT lines highlighting the importance of TREM2/DAP12 signaling in phagocytosis of this ligand.

To investigate whether the observed decrease in phagocytosis is specific to disease-relevant substrates or instead due to a more global down-regulation of phagocytic activity, we also tested phagocytosis of zymosan A, a dectin 1/2 agonist (Fig. 3e). This control ligand was taken up equally well by WT and TREM2 knockout cells, and phagocytosis of zymosan A was not altered by SYK inhibition, demonstrating that loss of TREM2 signaling results in substrate-specific deficits in phagocytosis (Fig. 3e).

**TREM2 antibody stimulation partially alters phagocytosis.** To further explore phagocytic deficits in TREM2 KO lines, we pre-stimulated isogenic microglia with anti-TREM2 antibody or an isogenic IgG control before exposing cells to APOE3,

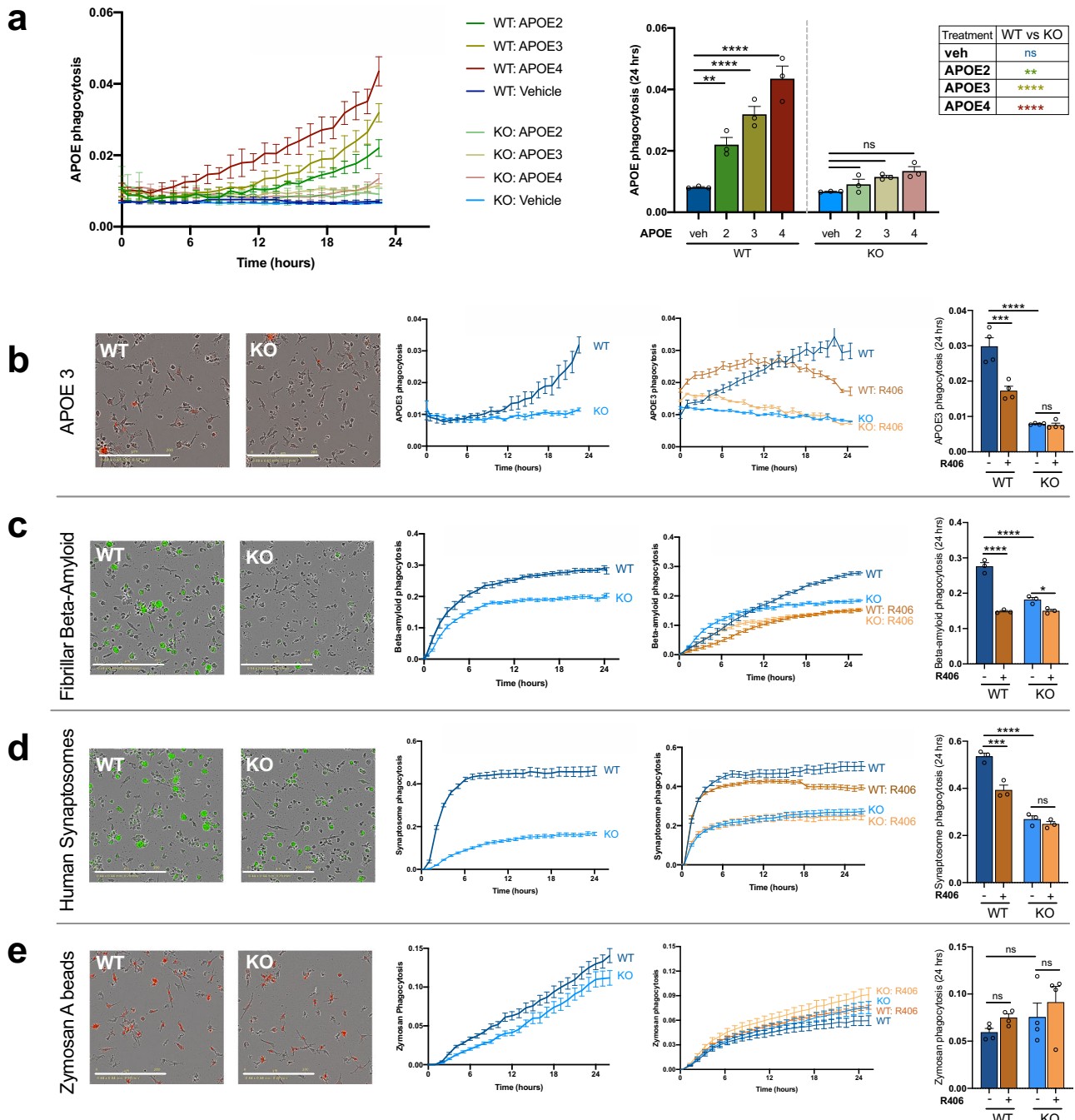

**Fig. 3 TREM2 knockout decreases phagocytosis of disease-relevant stimuli. a** Isogenic TREM2 WT and KO microglia were exposed to recombinant APOE 2 (green), APOE 3 (yellow), APOE4 (red), or a vehicle control (blue). Images were taken every hour for 24 h with IncuCyte S3 live imaging system. Scale bar: 200 μm. Statistical differences were quantified at 24 h (right, $n = 3$ independent wells with 4 images per well. Table (right) shows difference between WT and KO lines for each APOE genotype replicated in 2 isogenic backgrounds; two-way ANOVA, Tukey post hoc test, multiple comparisons. WT(veh) vs WT(APOE2): $p = 0.0052$, WT (APOE2) vs KO (APOE2): $p = 0.0075$, ****$p < 0.0001$, ns: $p > 0.8$.) Data are represented as mean values ± SEM. Microglia were exposed to **b** recombinant APOE3 (red), **c** fibrillar fluorescent beta-amyloid (green), **d** pHrodo-labeled human synaptosomes (green), or **e** pHrodo-labeled Zymosan A (red). Left shows representative images at 24 h. Inner graphs show untreated cells and the relative effects of TREM2 deletion alone on phagocytosis, or with addition of a SYK inhibitor (5 μM R406). Statistical differences were quantified at 24 h (right). For all panels, $n =$ 3–4 independent wells with 4 images per well; two-way ANOVA, Tukey post hoc test, multiple comparisons. **b** ***$p = 0.0002$, ****$p < 0.0001$, ns $p = 0.998$ **c** *$p = 0.0393$, ****$p < 0.0001$ **d** ***$p = 0.0008$, ****$p < 0.0001$, ns $p = 0.7912$ **e** WT vs WT + R406 $p = 0.301$, KO vs KO + R406 $p = 0.320$, WT vs KO $p = 0.344$. Data are represented as mean values ± SEM. Experiments were replicated in three isogenic lines with equivalent results.

beta-amyloid, synaptosomes, or Zymosan A (Supplementary Fig. 3). As expected, anti-TREM2 antibody treatment on TREM2 knockout lines did not have any effect. For phagocytosis of APOE3 and human synaptosomes, treatment of WT cells with anti-TREM2 antibody decreased phagocytosis. This decrease may be due to internalization of TREM2 following antibody stimulation that could, in turn, temporarily mimic a loss of function phenotype in response to secondary stimulation. With beta-amyloid and zymosan A, anti-TREM2 antibody had no effect, suggesting the TREM2 receptor itself may not be directly responsible for initiating phagocytosis of these substrates. However, in the case of beta-amyloid, we do show SYK signaling is important (Fig. 3c). This finding fits with the prior identification of several other microglial receptors that have been implicated in beta-amyloid internalization by microglia[41,42]. Additionally, this may suggest that the reduction of beta-amyloid phagocytosis in TREM2 knockouts may be more strongly influenced by changes in SYK activation state than by direct binding between TREM2 and beta-amyloid.

**TREM2 deletion reduces clustering around amyloid plaques and impairs migration toward beta-amyloid producing cultures.** Several groups have previously shown that TREM2 deficient mouse microglia exhibit decreased clustering around beta-amyloid plaques[24,43]. To determine whether human TREM2 knockout microglia exhibit a similar impairment, we transplanted GFP- or RFP-expressing isogenic human microglia into the brains of 5x-MITRG mice (Fig. 4a). These transgenic mice, obtained by backcrossing the 5x-fAD mouse model of AD with MITRG xenotransplantation-compatible mice (hCSF1, hCSF2, hTPO, Rag2 knockout, il2rγ knockout), were specifically developed to examine the functional behavior of transplanted human microglia in vivo[40]. Using this approach, we co-transplanted combinations of WT and TREM2 knockout human microglia (either GFP:WT, RFP:TREM2 KO or vice versa) into post-natal day 2–3 mice which were allowed to age for 6 months. Brain sections were stained with Amylo-Glo, a Thioflavin S analog, to detect fibrillar beta-amyloid plaques. As expected, WT human microglia exhibited a robust response to beta-amyloid plaque pathology, surrounding the plaques in a manner highly similar to that observed in human AD tissue[43]. In stark contrast, TREM2 knockout human microglia appeared unresponsive to plaque pathology, exhibiting little to no association with plaques and lacking the characteristic morphological changes observed in isogenic wild type microglia (Fig. 4a). These data are highly consistent with the response of murine TREM2 knockout microglia to plaque pathology and with observations in human TREM2 R47H cases[24,44,45], but also represent the first report to our knowledge that examines the impact of TREM2 deletion on human microglial plaque association.

Since, in vivo, beta-amyloid plaques are quite complex, consisting of not only beta-amyloid itself but many other proteins, dystrophic neurites, and reactive astrocytes, we next performed in vitro migration assays to determine if TREM2 knockout cells exhibit impaired migration toward beta-amyloid alone or whether additional signals derived from AD neurons might further influence the association between microglia and plaque pathology. To this end, we used two-chamber microfluidic migration devices[46] to measure the migration of WT and KO microglia toward soluble beta-amyloid or toward beta-amyloid producing human neuronal/glial mixed cultures. Whereas no significant differences were observed between WT and KO microglia in migration toward $A\beta_{1-40}$ or $A\beta_{1-42}$ alone, the more physiological combination of $A\beta_{1-40}$ and $A\beta_{1-42}$ revealed a significant impairment in the migration of TREM2 knockout microglia (Fig. 4b).

To determine whether additional neurally-derived cues may influence TREM2-mediated chemotaxis, we also analyzed microglial migration toward $A\beta$ producing human neurons and astrocytes plated within the central chamber. With co-culture of young (3-week old) neurons and astrocytes, we observed minimal microglial recruitment, although an impairment in TREM2 knockout migration was already evident (Fig. 4c). However, when neurons and astrocytes were aged to 9-weeks, a timepoint that was previously shown to correspond to increasing amyloid pathology and cell death in this system[46], we observed significant migration of WT microglia to the central chamber, whereas TREM2 knockout microglia remained completely unresponsive to these cell-derived chemotactic signals (Fig. 4c).

An outstanding question is whether the observed lack of migration is due to an inability of TREM2 deficient cells to sense and respond to chemoattractive cues or merely an inability of TREM2 knockout cells to move. To address this issue, we performed a scratch wound assay to observe general motility of TREM2 WT and KO microglia. This experiment showed no significant differences in baseline motility for WT and TREM2 knockout microglia suggesting that the differences quantified above are specific to chemoattractive migration (Fig. 4d).

**CXCR4 mediates migration deficits in TREM2 knockout microglia.** To further investigate the mechanism of impaired migration in TREM2 KO microglia, we returned to our RNA-sequencing data which revealed that expression of CXCR4, an important chemoattractive receptor, was reduced in the TREM2 knockout cells (Supplementary Data 1,3). Immunofluorescent and flow cytometry analysis of isogenic WT and TREM2 knockout microglia further confirmed that this receptor is decreased at the protein level in TREM2 knockout lines (Fig. 5a, b). Of further interest, CXCR4 expression has been shown to increase in murine disease associated microglia (DAMs)[30,47] and in our own recent study of plaque-associated human microglia[26], further supporting the potential role of this receptor in microglial migration toward plaques. CXCR4 has also been well characterized as a chemokine receptor in the peripheral immune system[48], responding to the ligand SDF-1α (CXCL12). CXCR4 is a canonical G-protein coupled receptor (GPCR) and its engagement results in the activation of $G_{q/11}$ and subsequent elevation of cytosolic $Ca^{2+}$ via $IP_3$ mediated ER-store release, a downstream response that is also implicated in TREM2 signaling. In the brain, SDF-1α is highly expressed in neurons and astrocytes particularly in the hippocampus[49,50] and increasing SDF-1α promotes the recruitment of microglia to plaques in AD mice[51].

To examine the potential role of CXCR4 signaling in iPS-microglia, cells were loaded with Fluo-4 and Fura-Red for ratiometric $Ca^{2+}$ imaging and treated with SDF-1α in $Ca^{2+}$-free buffer to unambiguously isolate $Ca^{2+}$ signals downstream of CXCR4 activation. Using this approach, we found that WT microglia showed significantly greater responses to SDF-1α than TREM2 knockout microglia (Fig. 5c, left and middle panels). Analysis of individual-cell responses further revealed that only a subset of cells were activated by SDF-1α in both WT and TREM2 knockout microglia (Fig. 5c right and Supplementary Fig. 4, Supplementary Movie 1, 2), suggesting that only a subset of microglia were primed to respond to SDF-1α signaling, and this percent was significantly lower in TREM2 knockout microglia (WT; 39% responded, KO: 11.5% responded). Single-cell quantification showed that the average peak $Ca^{2+}$ response to SDF-1α was also lower in TREM2 knockout microglia (Fig. 5c and Supplementary Fig. 4). These results strongly suggest that TREM2 knockout microglia have lower sensitivity to SDF-1α, likely due to the observed lower expression of CXCR4 mRNA and

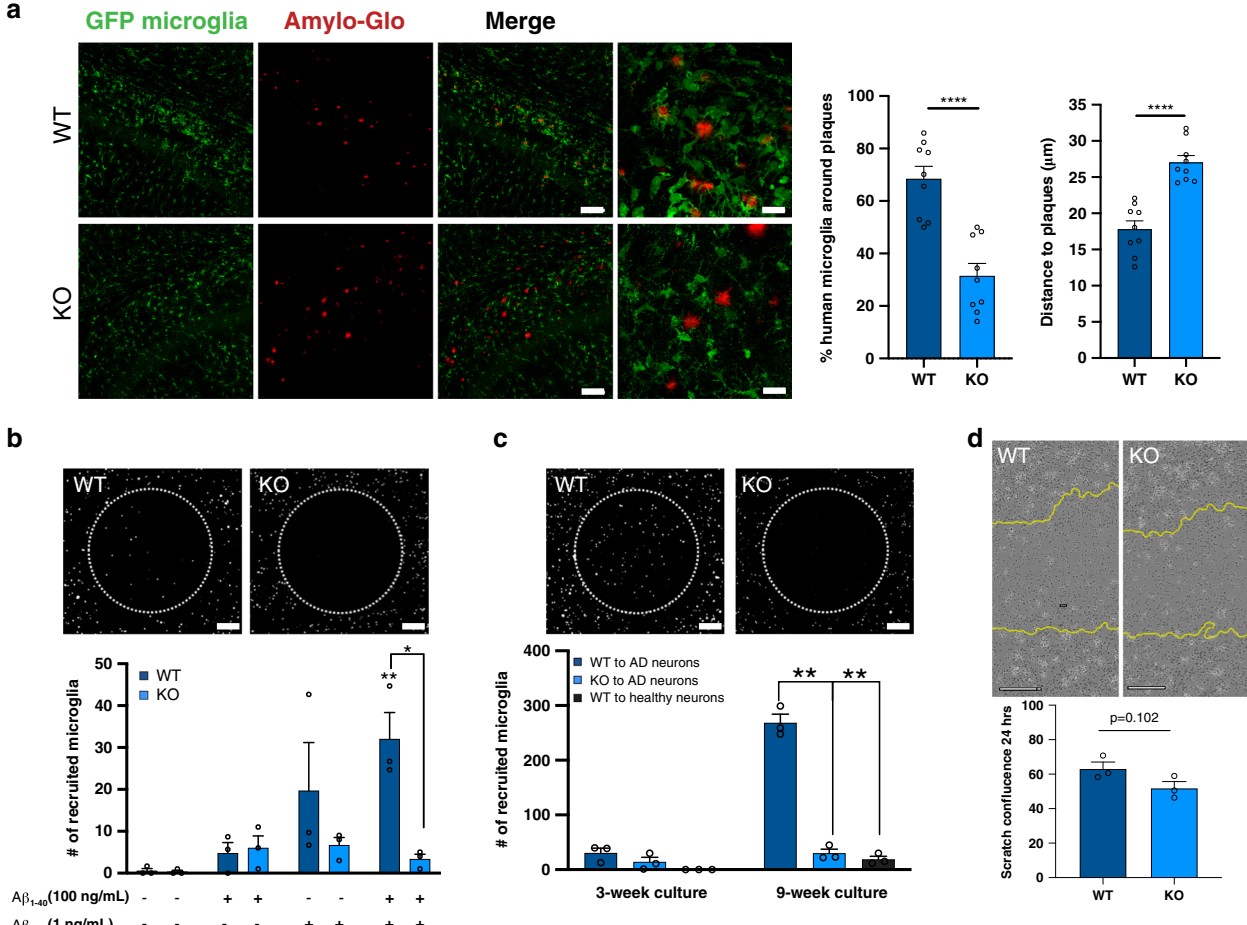

**Fig. 4 Deletion of TREM2 reduces the association of microglia with amyloid plaques and impairs migration toward amyloid and AD model cultures.**
**a** GFP-expressing TREM2 WT (top) or KO (bottom) microglia xenotransplanted into 5x-MITRG mice and aged 6 mo. were examined to assess the proximity between beta-amyloid plaques (red, Amylo-glo) and microglia (green). Scale bar low power: 40 μm; high power: 20 μm. Percent of each genotype within 50 μm of a plaque and raw distance to closest plaque was quantified (t-test ****$p < 0.0001$, $n = 9$ individual mice, 4 images per mouse). Darker blue represents TREM2 WT. Data are represented as mean values ± SEM. Experiment was run with two individual patient backgrounds. **b** In vitro migration of microglia toward soluble recombinant beta-amyloid. Images show microglia plated in outer chambers and allowed to migrate for 4 days through microfluidic channels toward beta-amyloid (A$\beta_{1-40}$ and A$\beta_{1-42}$) within the inner chamber (delineated by the dashed circle). Scale bar: 500 μm. ($n = 3$ independent devices, unpaired t-test *$p = 0.0114$; WT vs WT + A$\beta_{1-40}$ and A$\beta_{1-42}$ $p = 0.0076$). Data are represented as mean values ± SEM. **c** WT and TREM2 knockout microglial migration toward 3-week old Aβ-producing human neural and astrocyte mixed cultures or toward 9-week old wild type cultures. Scale bar: 500 μm. ($n = 3$ independent devices unpaired t-test **$p < 0.001$; WT vs KO $p = 0.0078$; WT vs WT(AD) vs WT (healthy) $p = 0.0063$). Data are represented as mean values ± SEM. Experiments from b,c were reproduced with equivalent results in two independent lines. **d** Scratch wound assay imaged and quantified 24 h post scratch with IncuCyte WoundMaker revealed no significant differences in general motility ($n = 3$ independent wells with 4 images per well, t-test $p = 0.102$). Yellow line demarcates original scratch. Data are represented as mean values ± SEM. Experiment was reproduced in three independent lines. Scale bar: 200 μm.

membrane-localized CXCR4 protein, which could, in turn, explain their migratory defect.

Next, to determine if CXCR4 signaling is necessary for migration, we used AMD3100 to block CXCR4 receptor activity during migration toward astrocyte and neural cultures[52]. Previous data highlights that this mixed model of neurons and glia does indeed express SDF-1α, and additionally, SDF-1α was shown to be increased when using Aβ-producing neurons and astrocytes compared to healthy controls[46]. We observed migration of WT microglia only in cultures in which SDF-1α was induced (AD neurons) and CXCR4 signaling was functional (no AMD3100, Fig. 5d). These data strongly suggest that CXCR4 is necessary for the migration of human microglia toward human beta-amyloid expressing neural and astrocyte cultures. Thus, these data support the possibility that activating CXCR4 in TREM2 deficient microglia may be a useful approach in AD to

rescue microglial migration toward dying cells and/or amyloid plaques.

**TREM2 knockout microglia fail to respond appropriately to amyloid plaque pathology in vivo.** It has been shown that expression of TREM2 is increased in the DAM sub-population found in AD mice[47], and is required for transition toward this phenotype[30,47]. However, recent data from our lab and others[26,53] suggests that the genes associated with human DAM formation have limited overlap with previously defined mouse datasets[54]. To determine if TREM2 KO effects DAM formation in human microglia, we co-transplanted iPS-derived hematopoietic progenitor cells into post-natal day 2–3 MITRG mouse pups allowing for the direct comparison of WT and KO human microglia within the same mouse. In order to identify which microglia were

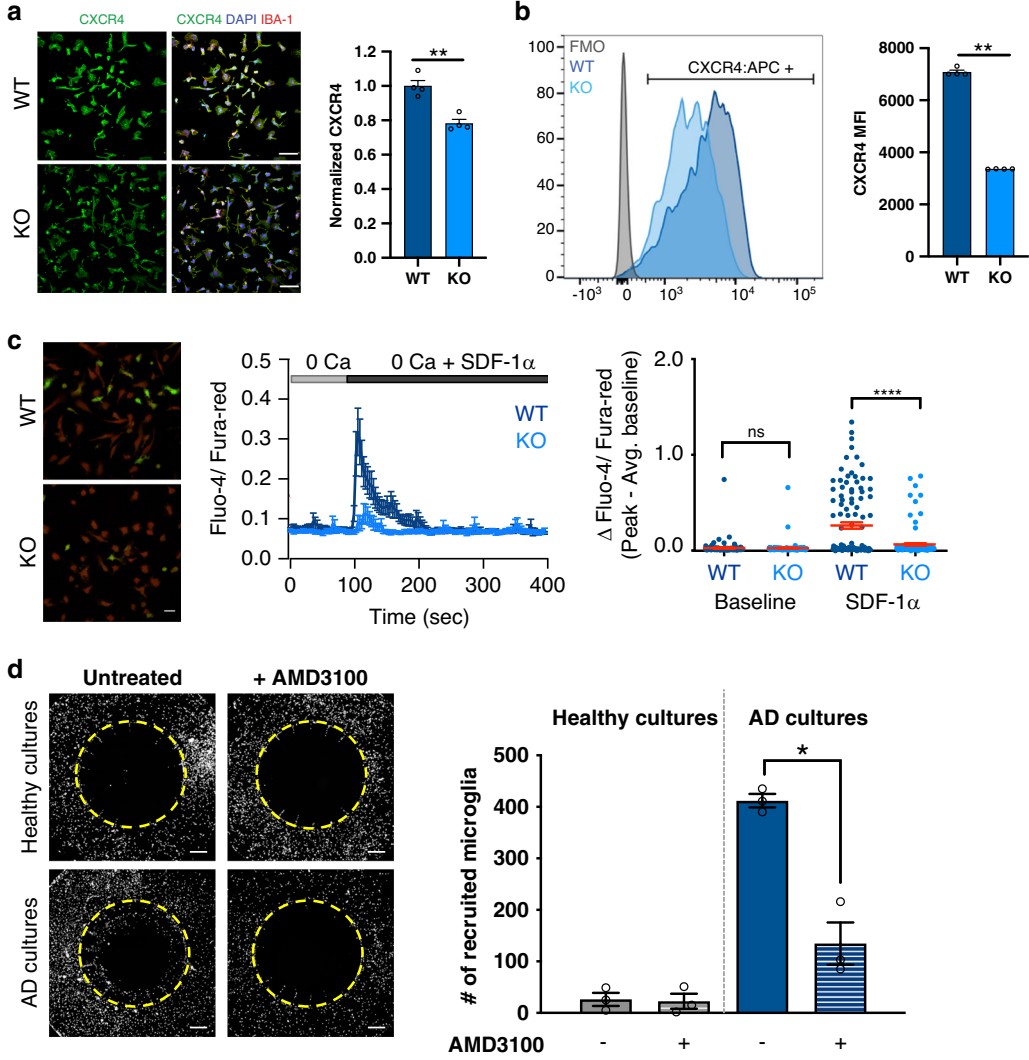

**Fig. 5 TREM2 knockout microglia are deficient in CXCR4 which is required for migration. a** Expression of CXCR4 (green), IBA-1 (red) in in vitro TREM2 isogenic lines. CXCR4 signal was normalized to DAPI (blue) intensity. Scale bar: 40 μm. In quantification, darker shades represent TREM2 WT. ($n = 4$ independent wells, 4 images per well, Experiment replicated in three independent lines; unpaired $t$-test $p = 0.0015$). Data are represented as mean values ± SEM. **b** Flow cytometry of CXCR4:APC (gated on fluorescence minus one (FMO) control (gray)) and quantification of mean fluorescence intensity (MFI) ($n = 3$ independent samples (100,000 events each), unpaired $t$-test $p = 0.0017$). Data are represented as mean values ± SEM. Gating strategy in Source data file. Reproduced with all three isogenic sets with similar results. **c** Overlay of maximum intensity projection images over time of Fluo-4 (green) and Fura-Red (red) loaded WT and TREM2 KO cells after activation with SDF-1α (left panel, Scale bar: 20 μm). Time-lapse run showing average cytosolic $Ca^{2+}$ response to 250 ng/mL SDF-1α measured by ratiometric Fluo-4 and Fura-Red signal (middle panel, Data are mean ± SEM; $n = 51$–61 cells). Summary of single cell baseline and maximal SDF-1α induced $Ca^{2+}$ elevation in WT and KO microglia (right panel, Data are mean ± SEM, $n = 111$–120 cells, 2 experiments). Y-axis denotes either peak baseline or peak SDF-1α response subtracted from the average baseline for each cell (****$p < 0.0001$, n.s not significant, as measured by One-way ANOVA; Post-hoc Tukey's multiple comparisons test). **d** WT microglia were allowed to migrate to 9-week old healthy or beta-amyloid producing (AD) neural/glial cultures plated within the central chamber (white dashed circle). Microglia pseudocolored gray. AMD3100 was used at 10 ng/mL. Scale bar: 50 μm (unpaired $t$-test, $p = 0.003$) Experiment was reproduced in two lines (total $n = 5$). Data are represented as mean values ± SEM.

derived from TREM2 WT versus knockout iPSCs, we mixed RFP-expressing WT cells with GFP-expressing KO cells and vice versa. Using this technique, we find long-term engraftment of human microglia within the mouse forebrain[26]. When crossed to the 5xfAD mouse (5x-MITRG) we also find that these cells respond to the onset and progression of disease with characteristic morphological and transcriptional changes.

After isolating human microglia from MITRG or 5x-MITRG mouse brains at 6 months of age, we performed single cell RNA sequencing to visualize the sub-populations of human microglia. In non-diseased MITRG mice we found four transcriptionally distinct sub-populations of microglia (Fig. 6a and Supplementary Data 4, 5, Supplementary Table 1). By analyzing the top genes in each population which significantly differ from the entire dataset, we determined that these four sub-populations were differentiated by genes involved in major histocompatibility complex (MHCII) and human leukocyte antigen (HLA) presentation (34.2%), the type 1 interferon response (10.8%), a small vaguely defined cluster which we termed "degranulation" (1.9%) and a "homeostatic" cluster that expressed high levels of canonical and homeostatic microglia genes while lacking expression of genes defining the other clusters (53.1%) (Fig. 6a and Supplementary Data 5) (clustering heatmaps and key gene expression UMAPs are included in Supplementary Fig. 5). However, when looking at

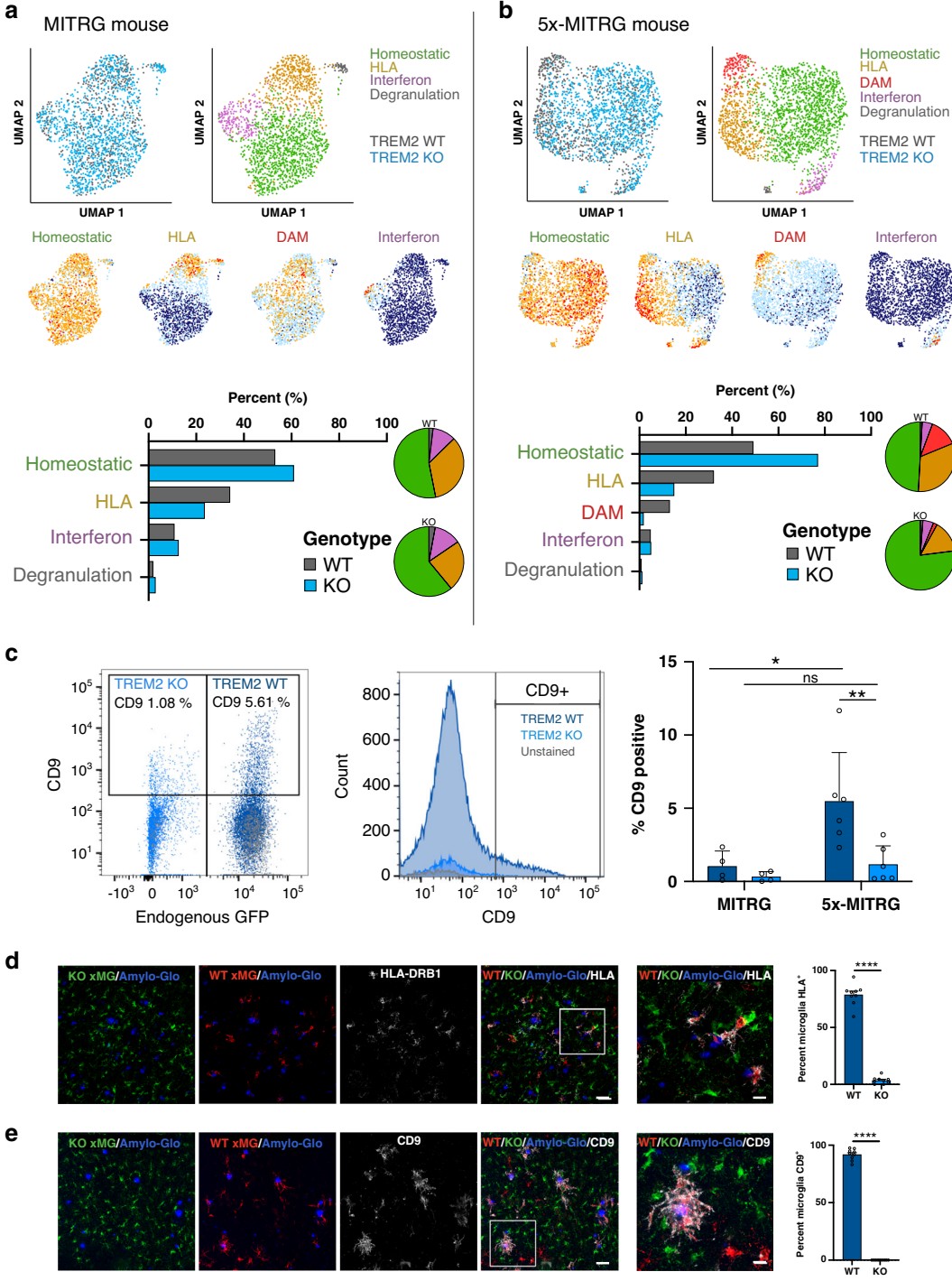

TREM2 knockout microglia transplanted into the same mouse, we saw these population percentages shift toward a more homeostatic profile even in the non-diseased mouse (Fig. 6a and Supplementary Table 1; 61% homeostatic, 23.5% HLA, 12.6% interferon, and 2.9% degranulation), consistent with the notion that TREM2 deletion may trap microglia in a homeostatic state[30] not only in disease but also in response to normal aging as well. This suggests that TREM2 knockout microglia are intrinsically hypo-reactive.

Next, we examined WT and TREM2 knockout microglia that had been co-transplanted into littermate 5x-MITRG mouse pups and allowed to age for 6 months. When WT microglia transplanted into this AD model were examined, we uncovered

five transcriptionally distinct sub-populations (Fig. 6b). The four populations seen in non-AD mice were still present as well as an additional population (13% of WT cells) that matches the human DAM population that we recently described[26]. As expected, exposure to beta-amyloid pathology, shifted WT microglia away from homeostatic and interferon high phenotypes, toward DAM phenotypes (Fig. 6a, b bar and pie graphs, Supplementary Table 1, WT and 5x analyzed together in Supplementary Data 5).

However, when TREM2 knockout microglia transplanted into the AD model were examined, we found that these cells fail to properly activate with 77% of TREM2 knockout microglia in the AD mice remaining homeostatic and only 1.8% transitioning toward the DAM subtype (~7 fold less than seen in WT cells,

**Fig. 6 Deletion of TREM2 suppresses the development of disease-associated microglia (DAMs) in vivo.** UMAP plots from WT and TREM2 knockout microglia transplanted into a **a** MITRG or **b** 5x-MITRG mouse. Top left shows the TREM2 genotype of each cell in the plot (gray = WT, blue = TREM2 KO) and the adjacent UMAP shows the clustering of human microglia sub-populations in non-diseased mice. Pie charts highlight the relative distribution of each TREM2 genotype within each cluster. The lower four UMAP plots demonstrate the relative expression of known homeostatic (green), human leukocyte antigen (HLA, yellow), interferon (pink), and DAM (red) markers. Homeostatic: CX3CR1, P2RY12, P2RY13, TMEM119, SALL1. HLA: HLA-DRA, HLA-DRB1, HLA-DRB5, HLA-DPA1, HLA-DPB1, HLA-DMA, HLA-DQA1, HLA-DQA2, HLA-DQB1, CD74. Interferon: IFIT1, IFIT2, ISG15, IFI6, IFITM3, MX1, MX2, STAT1. DAM: CD9, TREM2, SPP1, ITGAX, CD83, APOC1, LGALS3. Bar graphs show relative cluster percentages for each cell type. **c** Flow cytometry of co-transplanted TREM2 WT and KO microglia. Left dot plot shows GFP positive WT microglia (and GFP negative KO microglia) expression of CD9. Middle plot shows the same data as a histogram pre-split on RFP/GFP expression. As in (**a**), expected % CD9+ cells is around 5–10%. Right plot shows the quantification of all animals $n = 6$ independent 5x-MITRG mice and 4 MITRG mice. (two way ANOVA with Tukey post hoc test, $*p = 0.018$; $**p = 0.0099$, ns $p = 0.916$). Data are represented as mean values±SEM. Gating strategy in Source data file. **d** Histological analysis of human microglia within the 5x-MITRG mouse confirms TREM2 WT microglia (RFP+) express higher levels of the activation markers **d** HLA-DRB1 (gray) and **e** CD9 than TREM2 knockout microglia (GFP+). Reverse permutation of WT/KO shown in Supplementary Fig. 6. Scale bar low power: 40μm high power: 10μm ($n=9$ individual mice per genotype with 4 images per mouse. unpaired t-test two-tailed $****p < 0.0001$). Transplants were completed with one isogenic set. Data are represented as mean values±SEM.

Supplementary Data 5 and Supplementary Table 1). We additionally note that TREM2 KO microglia show ~2 fold less enrichment of the HLA high cluster. Thus, human TREM2 knockout microglia exhibit a similar impairment in the response to beta-amyloid pathology as murine microglia, although the precise gene set of this lost 'DAM' population only partially overlaps (~12%) with that of the murine counterpart[26].

**TREM2 knockout microglia do not form DAMs in vivo.** Because our single-cell sequencing was performed with only one animal per genotype to provide a discovery-based approach and because RNA does not always directly correlate with protein expression[55], next we sought to corroborate our results by examining protein expression in a larger cohort of mice. To validate our findings that CD9 expression (marker of the DAM cluster[26]) primarily increased in WT cells (not TREM2 KO) in the AD model brain (not WT) we performed flow cytometry for CD9 in all 4 sets of animals (MITRG + WT, MITRG + KO, 5x-MITRG + WT, 5x-MITRG + KO). Again, we used co-transplanted animals and thus cells from the same brain could be separated by GFP or RFP expression to denote TREM2 genotype (Fig. 6c, left). This analysis confirmed that non-diseased animals (MITRG) express very low levels of CD9 protein. We also confirm that CD9 levels are significantly higher for WT cells in an AD-model brain (5x-MITRG) than KO cells from the same brain. Indeed, there is no significant difference in the number of CD9 positive "DAMs" for KO cells in the non-diseased and AD-model brains (Fig. 6c). This, again, suggests that TREM2 KO cells are locked in a homeostatic state despite exposure to beta-amyloid pathology. Since these cells are isolated from co-transplanted brains of WT and KO cells, these results also suggest that the presence of activated WT cells is not sufficient to induce a DAM phenotype in nearby KO cells.

To further validate and confirm our single-cell sequencing and flow cytometry results at the anatomical level, brain slices from the cortex of the same co-transplanted mice were examined. Expression of HLA and DAM markers increase specifically around beta-amyloid plaques, consistent with our recent publication[26]. We highlight that TREM2 knockout microglia (GFP) do not exhibit the typical plaque-associated increase in expression of HLA markers (Fig. 6d, HLA-DRB1) or DAM markers (Fig. 6e, CD9) as seen with WT cells (RFP), consistent with the notion that TREM2 knockout cells are locked into a more homeostatic phenotype. Importantly, mice transplanted with the opposite combination of GFP/RFP cells show an equivalent loss of CD9 and HLA-DRB1 in TREM2 KO (RFP) microglia (Supplementary Fig. 6). Together, these data robustly

show that human TREM2 knockout microglia fail to activate in response to beta-amyloid plaques in vivo.

## Discussion

By combining TREM2 stimulation and RNA sequencing, we were able to improve our understanding of the downstream genetic targets of TREM2 signaling (Fig. 1). By using an anti-human TREM2 antibody to specifically stimulate TREM2 downstream signaling cascades we have uncovered TREM2-specific transcript alterations that were enriched for gene sets related to cellular migration and survival. Because these experiments were carried out using human cells and stimulation of human TREM2, our RNA-sequencing data may also prove useful to investigators seeking to better understand the potential therapeutic application of TREM2 stimulating antibodies that have recently began testing in early stage clinical trials.

Consistent with our sequencing results with functional experiments, we have highlighted viability differences in our TREM2 isogenic microglia. Even without any stimulation or stress, TREM2 knockout microglia show lower viability (Fig. 2b). When additional stressors such as loss of growth factors are added, this phenotype is exacerbated (Fig. 2a, c). Furthermore, we confirm that depletion of CSF1R signaling alone is sufficient to replicate this response.

Guided by our sequencing results, we also examined microglial phagocytosis of APOE (Fig. 3a, b). Previous studies examining how APOE isoforms interact with TREM2 have been inconsistent. For example, Atagi et al. found that TREM2 binds all APOE isoforms with similar affinity, whereas Bailey et al. found that TREM2 showed the highest affinity for APOE4 above APOE3 or APOE2[56,57]. Here, we were able to demonstrate that APOE variants are recognized and engulfed by human microglia at different rates with APOE4 > APOE3 > APOE2. However, whether or not this result contributes to the known effect of APOE on disease risk remains unclear. Of further interest, we find that microglia lacking TREM2 do not internalize any APOE. This occurs despite normal expression of canonical APOE receptors (no difference from WT lines by RNA-sequencing). Together, this data suggests that TREM2 is necessary for microglial responses to APOE and may even be the only functional APOE receptor on human microglia.

We have also investigated microglial responses to other disease-relevant stimuli through phagocytosis. We show that TREM2 knockout microglia perform less phagocytosis of human synaptosomes and beta-amyloid but not zymosan A and that this phenotype occurs partially via decreased signaling through SYK in KO cells (Figs. 1, 3). While dectin-1 stimulation (by zymosan A) does induce phosphorylation of SYK, it has been shown that

the resulting phagocytosis is independent of SYK[58,59]. We corroborate this data by showing that phagocytosis of zymosan A is unchanged by the presence of SYK inhibitor R406 (Fig. 3e). Thus, we conclude that TREM2 knockout lines are deficient only in SYK-dependent phagocytosis. This pathway makes intuitive sense given that TREM2 signal transduction includes phosphorylation of SYK (Fig. 1f, g), which we suggest is necessary for a normal phagocytic response in these disease-relevant conditions.

While decreased synaptic phagocytosis in TREM2 knockout microglia (modeled here with synaptosomes) may be surprising given that synaptic over-pruning is hypothesized to worsen AD progression, similar results have been shown in murine microglia[39,60]. Additionally, it is possible that the ligand-binding domain TREM2 mutations most strongly associated with AD, may not alter the recognition of this specific phagocytic substrate in the same way as a loss-of function deletion. Additionally, TREM2 function in synaptic pruning may be isolated to developmental stages in vivo. It has recently been shown in mice that TREM2 knockout results in hyper-connectivity in the brain and indeed lower expression of TREM2 is correlated with increased prevalence of autism spectrum disorder[39].

Based on our RNA sequencing results and previous research in the field, we next investigated the role of TREM2 in motility and directed migration. By combining studies of human microglia both in vitro and in vivo, we conclude that TREM2 knockout microglia fail to migrate toward beta-amyloid as well as beta-amyloid producing neurons and astrocytes (Fig. 4). We also report, that expression of the chemoattractive receptor, CXCR4, is diminished in TREM2 knockout microglia and exposure to its ligand, SDF-1α, uncovers a significant deficit in the calcium response to this ligand, suggesting that loss of CXCR4 signaling likely contributes to the impaired migration of TREM2 knockout microglia to beta-amyloid plaques (Fig. 5). To corroborate these data, we show that blocking CXCR4 signaling with AMD3100 is sufficient to prevent migration of WT microglia toward beta-amyloid producing neurons and astrocytes (Fig. 5d). This suggests that CXCR4 signaling may offer a potentially useful therapeutic target to re-mobilize microglia toward amyloid plaques and degenerating neurons.

Of interest, SDF-1α as well as other endogenous CXCR4 ligands, including ubiquitin and macrophage migration inhibitory factor (MIF) have been shown to be increased in AD brain tissue particularly around plaques, co-localizing with dystrophic neurons and within murine DAM populations[61–64]. This further highlights the need for exploration of the role of CXCR4 in microglial migration in disease particularly as a potential therapeutic for patients with mutations in TREM2.

Lastly, because microglial function and gene expression are greatly influenced by other cell types and environmental factors in the brain, we performed xenotransplantation of human microglia into the murine brain. Previous studies have shown that activation of microglia with a strong pro-inflammatory stimuli (e.g., LPS) decreases TREM2 expression hinting that levels of TREM2 may be a crucial indicator of microglial activation[65]. Conversely, TREM2 expression increases in the DAM subpopulation found in AD mice[47]. This microglial profile was delineated by Keren-Shaul et al. as well as Krasemann et al. who showed that TREM2 expression is required for transition to this disease-responsive transcriptome signature. However, recent data from our lab shows that the DAM prolife in human microglia is distinct from those shown in Keren-Shaul and Krasemann's murine DAMs[26], thus bringing into question whether human TREM2 similarly prevents human DAM formation in vivo. We, therefore, performed xenotransplantation of human TREM2 isogenic microglia and were able to examine responses of WT and TREM2 KO microglia in both AD-model and non-diseased mice.

This investigation allowed us to demonstrate that TREM2 knockout microglia are generally locked in a homeostatic state and resistant to adopt a DAM transcriptomic state. Interestingly, we note that this is the case not only in AD model mice but also in non-diseased states as well, suggesting that deficits in TREM2 signaling intrinsically affect human microglia responses, though this effect is greatly exacerbated in disease.

These experiments replicated the same human microglial clusters published in Hasslemann et al. 2019[66] and reveal that TREM2 is partially required for transition into the HLA and DAM clusters in disease. Loss of TREM2 significantly increases the percentage of microglia present in the homeostatic cluster at the expense of these clusters (Fig. 6), which we confirmed by both flow cytometry and immunohistochemistry. Whether or not this result causes the migration deficit (lack of DAM activation produces a lack of migration) or occurs as a result of this deficit (lack of migration to plaques results in fewer DAMs) is not yet understood. However, we do see CXCR4 expression highly expressed in the human DAM population and decreased in TREM2 KO microglia (Fig. 5 and Supplementary Data 4), suggesting that TREM2 is required to elicit CXCR4-dependent migration toward plaques and the subsequent adoption of a DAM state.

Of further interest, depending on how we set clustering of the single-cell sequencing data, we uncovered an additional TREM2-dependent shift in a cluster of p53 apoptotic/senescent microglia (65% of genes associated with p53 including 40.5% of the population known to be direct targets of p53). These cells are likely either senescent or at the earliest stages of cell death, since fully dead cells are removed within the sequencing analysis pipeline (see "Methods"). This senescent cluster was only observed in 5x-MITRG samples and was enriched almost 10-fold in TREM2 knockout microglia (0.6% of WT cells, 5.8% of TREM2 knockout cells) (Supplementary Fig. 7). This finding closely parallels our in vitro data (Figs. 1e, i, 2) which has also implicated p53 signaling and showed that TREM2 knockout microglia are hypersensitive to stress, inducing apoptosis at a much higher rate than WT cells. However, we were unable to identify human-specific antibodies to confirm this population by histology and thus re-clustered our data to include only verified clusters. It is also possible that we do not see this cluster in vivo because it is an artifact of the isolation protocol, though that does not seem to be the entire explanation given that this cluster appears only in 5x-MITRG samples and not the non-diseased animals which were isolated in parallel. Thus, we posit that human TREM2 deficient cells exposed to Alzheimer's pathology may be more sensitive to isolation.

Together, the data in this manuscript highlight the utility of combining in vitro and in vivo analysis to produce complementary datasets in order to better understand the complex mechanisms of microglial biology. We present evidence of TREM2 involvement in CSF1R-dependent survival, SYK-dependent phagocytosis of synaptosomes, beta-amyloid, and APOE, CXCR4-dependent migration, and formation of the human DAM response. These molecular studies have elucidated microglial processes and receptors that may be important in the progression of AD. Future studies will be needed to understand which of these mechanisms underlie the risk incurred by loss of TREM2 function before they can be studied as potential therapeutic targets for the treatment of neurodegenerative diseases.

## Methods

**Animals**. All animal procedures were conducted in accordance with the guidelines set forth by the National Institutes of Health and the University of California, Irvine Institutional Animal Care and Use Committee, who approved the study protocol. The MITRG mouse was purchased from Jackson Laboratories (stock

#017711); this BALB/c/129 model includes two knockouts alleles, Rag2-(Rag2tm1.1Flv), γc- (Il2rgtm1.1Flv), and three humanized knock-in alleles, M-CSFh (Csf1tm1(CSF1)Flv), IL-3/GM-CSFh (Csf2/Il3tm1.1(CSF2,IL3)Flv), TPOh (Thpotm1.1(TPO)Flv). The related and parental M-CSFh mouse line was also purchased from Jackson Laboratories (stock # 017708) and contains Rag2 and Il2rg deletions and humanized M-CSFh. The 5xFAD-MITRG model was created by backcrossing the MITRG mouse with 5xFAD mice which overexpress co-integrated transgenes for Familial Alzheimer's Disease (FAD) mutant APP (Swedish, Florida, and London) and mutant FAD PS1 (M146L and L286V). Progeny of these pairings were then genotyped (Primers are provided in Supplementary Table 2) and backcrossed with MITRG mice to return the 5 MITRG genes to homozygosity and maintain the APP/PS1 transgenic loci in the hemizygous state, resulting in the 5xfAD-MITRG model (Rag2-; γc-; M-CSFh; IL-3/GM-CSFh; TPOh; Tg (APPSwFlLon,PSEN1*M146L*L286V)6799Vas). All mice were age and sex matched and group housed on a 12 h/12 h light/dark cycle with food and water ad libitum. Mice are housed with ambient temperature and humidity.

**Generation of iPSC cell lines from human fibroblasts**. Human iPSC cell lines were generated by the University of California, Irvine Alzheimer's Disease Research Center (UCI ADRC) Induced Pluripotent Stem Cell Core from subject fibroblasts under approved Institutional Review Boards (IRB) and human Stem Cell Research Oversight (hSCRO) committee protocols. Informed consent was received by each of the participants who donated fibroblasts. Non-integrating Sendai virus was used to perform reprogramming, thereby avoiding any integration-induced effects (Cytotune 2.0). iPSCs were confirmed to be karyotype normal by G-banding, sterile, and pluripotent via Pluritest Array Analysis and trilineage in vitro differentiation. The Pluritest is a microarray-based assessment of pluripotency based on iPS whole transcriptome analysis referenced to a library of functionally validated iPSCs (https://www.pluritest.org/). Additional GFP- and RFP-αtubulin expressing iPSC lines (AICS-0036 and AICS-0031-035) were purchased from Corriel and originally generated by Dr. Bruce Conklin. To avoid the potential selection of latent or hidden infections all iPSC culturing and microglia differentiation and experimentation is performed without the use of antibiotics. In addition, all lines are routinely tested for sterility including mycoplasma testing. iPSCs were maintained feeder-free on matrigel in complete TeSR-E8 medium (Stemcell Technologies) in a humidified incubator (5% $CO_2$, 37 °C). All lines will be available upon request to the corresponding author.

**Differentiation of iPS-microglia from iPSCs[27]**. iPS-microglia were differentiated as detailed in McQuade et al.[27]. Briefly, iPSCs are differentiated to hematopoetic progenitor cells using the STEMdiff Hematopoetic kit for 10–12 days before passage into microglia differentiation medium[27,40] including DMEM/F12, 2× insulin-transferrin-selenite, 2× B27, 0.5× N2, 1× glutamax, 1× non-essential amino acids, 400 μM monothioglycerol, and 5 μg/mL human insulin. Cultures were maintained in this basal medium supplemented with 100 ng/mL IL-34, 50 ng/mL TGF-β1, and 25 ng/mL M-CSF (Peprotech) for 28 days. For the last 3 days in culture, two additional cytokines were added (100 ng/mL CD200 (Novoprotein) and 100 ng/mL CX3CL1 (Peprotech)) to mature the microglia in a homeostatic brain-like environment[40].

**CRISPR-mediated knockout of Trem2 in iPSCs**. This manuscript uses four independent isogenic sets of TREM2 knockout iPSCs. These lines were each made on different patient iPSC backgrounds (two male, one female; all lines APOE3/3). $2 \times 10^5$ induced pluripotent stem cells were isolated following Accutase enzymatic digestion for 3 min at 37 °C. Cells were resuspended in 60 μL nucleofection buffer from Human Stem Cell Nucleofector™ Kit 2 (Lonza). The suspension was combined with 50 μg of RNP complex formed by incubating Alt-R® S.p. HiFi Cas9 Nuclease V3 (IDTDNA) with fused crRNA:tracrRNA (IDTDNA) duplex for 15 min at 23 °C. The suspension was transferred to the Amaxa Nucleofector cuvette and transfected using program B-016. Cells were plated in TeSR™-E8™ (STEMCELL Technologies) media with 0.25 μM Thiazovivin (STEMCELL Technologies) overnight to recover. Cells were digested the following day with Accutase and single-cell plated to 96-well plates in TeSR™-E8™ media with 0.25 μM Thiazovivin and CloneR™ (STEMCELL Technologies) supplement for clonal isolation and expansion. Genomic DNA was extracted using Extracta DNA prep for PCR (Quantabio) from a sample of each clone upon passage and amplified for sequencing using Taq PCR Master Mix (ThermoFisher Scientific) at the cut site (Primers are provided in Supplementary Table 2). PCR product from promising clones was transformed using TOPO™ TA Cloning™ Kit for Subcloning, with One Shot™ TOP10 (ThermoFisher Scientific) for allele-specific sequencing. The top three off-target sites as identified by IDTDNA were amplified and sequenced per clone of interest to confirm absence of off-targeting before banking for future experiments.

**Immunoblotting**. SDS/PAGE Western blot was performed according to previously established protocols[67]. Two human anti-TREM2 antibodies were used to confirm knockdown and knockout (R&D AF1828, Sigma HPA012571). Anti-P-SYK was probed with Cell signaling 2710 (1:1000); anti-SYK was probed with Cell Signaling 13198 (1:2000). Following initial labeling, blots were stripped in 0.2 N NaOH for 15 min. anti-GAPDH (47724 Santa Cruz 1:5000) was probed to control for protein

loading. Blots were visualized with Goat Anti-Mouse (Millipore, AP308P; 1:10,000) or -Rabbit (Millipore, 12-348; 1:10,000) HRP conjugate and on a Bio-Rad ChemiDoc Gel Imaging System. Images were quantified in ImageJ (version 2.0.0) and statistical analyses were carried out in GraphPad Prism 7.

**Homogenous time-resolved fluorescence (HTRF)**. Cisbio human TREM2 HTRF kit was used as per manufacturer's instructions. 100,000 iPS-microglia were plated in a 96-well format overnight in 200 μL microglia maintenance medium. Conditioned medium was collected and cells were lysed in Lysis Buffer 3. After antibody addition, plates were incubated overnight at room temperature and read on CLARIOstar imager (BMG Labtech).

**RNA isolation and sequencing preparation**. Total RNA was isolated using RNeasy Mini kit (Qiagen). One million iPS-microglia were lysed in 700 μL RLT buffer and RNA was isolated per the manufacturer's instructions with DNAse treatment (10 min, RT). Centrifugation times were increased to $16,000 \times g$ for 1.5 min to maximize yield. RNA integrity was measured using the Bioanalyzer Agilent 2100. All libraries were prepared from samples with RNA integrity values ≥9.5. 500 ng RNA per sample was used to create RNA-seq libraries through the Illumina TruSeq mRNA stranded protocol. Samples were sequenced on the NovoSeq S4 chip (WT vs KO Fig. 1c and neuron treatment Supplementary Fig. 1) or Illumina HiSeq 4000 platform (antibody treatment Fig. 1j).

**Cell death assay**. iPS-microglia were plated at 30 % confluence into a 96-well plate (4 wells per line per condition). At time 0, all microglia were treated with IncuCyte Caspase-3/7 Green Apoptosis Assay Reagent 1:1000. Cells were maintained in the described medium: fresh complete medium, fresh basal medium + 100 ng/μL IL-34 + 25 ng/μL M-CSF, fresh basal medium + 50 ng/μL TGF-B1, or basal medium with no cytokines for 3 days. Four 20× images per well were collected every hour. Using IncuCyte 2018B software, image masks for phase confluence (visually gated out apoptotic cells) as well as caspase 3/7 signal (green) were generated. Graphs display caspase normalized to phase confluence. Completed with 2 lines.

**Collection and pHrodo labeling of human AD synaptosomes**. Human brain tissue samples were obtained from the UCI ADRC from patients who have given informed consent. These samples were from deceased AD patients upon autopsy (PMI < 3 h) and slowly frozen and stored in isotonic 0.32 M sucrose, 10 mM HEPES, pH 7.4 at −80 °C. Synaptosome preparation was adapted from Prieto et al.[68]. Samples were thawed at 37 °C and homogenized using a pre-cooled glass/Teflon homogenizer (clearance 0.1–0.15 mm) with addition of protease inhibitors, phosphatase inhibitors (Thermo Scientific), and an antioxidant cocktail (Sigma-Aldrich; #A1345). Brain homogenate was centrifuged at $1200 \times g$ for 10 min at 4 °C and the resulting supernatant fraction (S1) was collected. The S1 fraction was centrifuged at $12,000 \times g$ for 20 min at 4 °C and the resulting supernatant (S2) was aspirated. The crude synaptosome pellet (P2) (containing synaptosomes and mitochondria) was labeled for 45 min at RT with amine reactive pHrodo™ Red SE or pHrodo™ Green STP ester (Thermo), a lipophilic, fluorogenic dye that increases in fluorescence as the surrounding environment acidifies. Labeled synaptosomes were washed twice with excess ice-cold 1× HBSS. For each washing step and experimental procedure, synaptosomes were resuspended by gently pipetting up and down using Finntip pipette tips (Thermo) to minimize shear force stress and preserve synaptosome integrity.

**Fluorescent APOE collection and validation**. Hek293T cells were cultured at 37 °C, 5% $CO_2$ in DMEM supplemented with 10% (vol/vol) FBS. Cells were plated at 250,000 cells/mL and transfected 24 h later with Lipofectamine 2000 and 1 μg of DNA for APOE2, 3, or 4 tagged with mCherry-SepHluorin. Media was changed 24 h after transfection, washed once with PBS, and incubated for 48 h in microglial basal medium. Media was collected and centrifuged at $750 \times g$ for 5 min and transferred to a new tube to eliminate dead cell debris.

To assess APOE concentration across isoforms, 20 μL of media was analyzed by SDS/PAGE on a NuPage Novex 4–12% Bis-Tris precast gel with MOPS running buffer (Invitrogen) and transferred onto an Immobilon-FL PVDF (LI-COR) membrane. Whole protein was quantified using the revert assay, and the membrane was blocked with Intercept (TBS) Blocking Buffer (LI-COR) for 1 h. The membrane was then incubated in primary antibody overnight, washed three times with TBS-0.1% Tween-20, and incubated for 1 h in Intercept block supplemented with 0.1% Tween-20 and near-infrared conjugated secondary antibody. Membranes were imaged on a LI-COR scanner and quantified using Empiria Software.

To assess the lipidation state of APOE in the media, 20 μL of conditioned media from APOE transfected Hek293T cells or 2 μL of HA-tagged APOE protein purified from bacteria (construct generated in-house) was added to 6.25 μL NativePAGE 4× Sample Buffer (Invitrogen BN2008) and run on a NativePAGE mini gel (Invitrogen BN2112BX10). NativePAGE Running Buffer (Invitrogen BN20001) was used to make light and dark cathode buffer and anode buffer. Gel was transferred onto an Immobilon-P PVDF membrane (Millipore IPVH00010). The membrane was blocked with Pierce StartingBlock Buffer (Fisher Scientific EN37543) for 30 min and incubated overnight with primary antibody (803301,

Biolegend, 1:1000), washed three times with TBS-0.1% Tween-20, and incubated for 1 h in StartingBlock Buffer with HRP-conjugated secondary antibody (115-035-146 Jackson Immunoresearch, 1:10,000). HRP was detected using SuperSignal West Dura Chemiluminescent reagents (Fisher Scientific PI34076) and imaged using an Azure c600 system.

**Phagocytosis assay**. Phagocytic activity of isogenic iPS-microglia was examined using the IncuCyte S3 Live-Cell Analysis System (Sartorius). Microglia were plated at 50% confluence on Matrigel-coated 96-well plates. Four fields of view in each of four wells were captured for each condition. 15 min after plating, 50 µL APOE containing medium, 50 µg/mL pHrodo tagged human AD synaptosomes, 100 ng/mL pHrodo tagged zymosan A beads, or 2 µg/mL fluorescent beta-amyloid was added to cells. Images of phase and fluorescence were captured in the IncuCyte S3 live cell imager. Using IncuCyte 2019B software, image masks for fluorescent signal (phagocytosis of each substrate) were normalized to cell body area. APOE phagocytosis completed with two lines.

**Ratiometric Ca$^{2+}$ imaging using Fluo-4 and Fura-Red**. iPSC-derived microglia were plated on fibronectin-coated 35 mm, No. 1.5 thickness glass bottom dishes (1:100, MatTek Corporation). After 24 h of culture, cells were loaded with 3 µM Fluo-4 AM and 3 µM Fura-Red AM (Molecular Probes) in the presence of Pluronic Acid F-127 (Molecular Probes) for 30 min at room temperature (RT). Ca$^{2+}$ dyes were then washed off 3 times using microglial cell culture media and cells were resuspended in Ca$^{2+}$ Ringer's solution for 10 min to allow de-esterification of the dyes. Time-lapse images were acquired at RT in a single z-plane at 20 frames/minute using Olympus Fluoview FV3000i confocal laser scanning microscope equipped with high speed resonance scanner, IX3-ZDC2 Z-drift compensator and a 40× silicone oil objective (NA 1.25). Fluo-4 and Fura-red were both excited using a 488 nm diode laser (0.05 % laser transmissivity, 10% laser ND filter) and two high-sensitivity cooled GaAsP PMTs set to wavelengths 494–544 nm and 580–680 nm were used for detection in the green and red channels respectively. Images were exported to Image J, background subtracted and single-cell analysis was done by drawing ROIs around each cell in the field. Average intensities in the green and red channels were calculated for each ROI at each time-point. Fluo-4/Fura-Red ratio was then obtained to further generate traces showing single-cell and average changes in cytosolic Ca$^{2+}$ over time. Completed with 2 lines.

**Creation and culture of human neural progenitor cells (NPCs)**. ReN cell VM human neural progenitor cells (hNPCs, EMD Millipore, Billerica, MA, USA) were transfected with commercially available APPSL-GFP Alzheimer's lentiviruses (EMD Millipore) to develop ReN cells producing high levels of Aβ (AD hNPCs) through overexpression of a variant of the human amyloid precursor protein (APP) containing K670N/M671L (Swedish) and V717I (London) FAD mutations (APPSL). As previously published[69], cells were transfected with 50–100 µL vital solution (1 × 10$^6$ TU/mL) and incubated for 24 h. After, cells were washed 3× to stop infection. Expression of infected genes is confirmed by fluorescence. For the control counterpart, ReN cells were transfected with the control GFP construct (LentiBrite™ GFP Control, EMD Millipore) to develop control hNPCs. After 3 days of incubation, the transgene positive cells were enriched by FACS sorting (BD FACS Aria II, BD Biosciences). The either of hNPCs or AD hNPCs was plated onto the culture flask coated with 1% Matrigel (BD Biosciences, San Jose, CA, USA) in DMEM/F12 (Life Technologies, Grand Island, NY, USA) media supplemented with 2 mg heparin (StemCell Technologies, Vancouver, Canada), 2% (v/v) B27 neural supplement (Life Technologies, Grand Island, NY, USA), 20 mg EGF (Sigma-Aldrich, St Louis, MO, USA), 20 mg bFGF (Stemgent, Cambridge, MA, USA), and 1% (v/v) penicillin/streptomycin/amphotericin-B solution (Lonza, Hopkinton, MA, USA) and incubated at 37 °C supplied with 5% CO$_2$. Cell culture media were changed every 3 days until cells were confluent.

**Preparation of 3D human AD models in the microfluidic device**. We employed our 3D organotypic microfluidic model mimicking pathological signatures of human AD brains, developed by Park et al.[46], to compare chemotaxis of iPS-microglia in response to AD cues. Briefly, ReN cells were 3D cultured in the central chamber of microfluidic device at the cell density of 2 × 10$^6$ cells/mL in the 10% of Matrigel diluted with DMEM/F12 differentiation media supplemented with 2 mg heparin, 2% (v/v) B27 neural supplement, and 1% (v/v) penicillin/streptomycin/amphotericin-B solution without growth factors. The microfluidic devices were incubated at 37 °C supplied with 5% CO$_2$ and replaced one half volume of the differentiation media every 3.5 days until the cells were fully differentiated into neurons and astrocytes (approximately 2.5 weeks). To develop early AD and late AD models, the cells were further incubated in the device for 0.5 weeks (3-week AD model) and 7.5 weeks (9-week AD model).

**Migration study of iPS-microglia in response to beta-amyloid**. iPS-microglia (2000 cells/device) were loaded in the angular chamber to test the activation and chemotaxis of microglia to the central chamber containing either Aβ$_{40}$ (100 ng/mL), Aβ$_{42}$ (1 ng/mL), 3-week AD model, 9-week healthy model, or 9-week AD model. To characterize motility, we monitored the number of recruited microglia in the

central chamber for 4 days under the fully automated Nikon TiE microscope (×10 magnification; Micro Device Instruments, Avon, MA, USA).

**Scratch wound assay**. Motility was observed using Essen Incucyte WoundMaker Assay. 100,000 iPS-microglia were plated per well on fibronectin-coated 96-well plate (n = 3 wells) for 1 h for cells to become adherent. Scratches were repeated 4×. Cells were imaged at 24 h post-scratch. Wound confluence (confluence of cells within original wound ROI) was calculated using Incucyte 2019B Software.

**Flow cytometry**. Microglia were treated with FC block (1:100; BD Biosciences 553142) 5 min in FACS buffer (DPBS, 2% BSA, 50 µM EGTA) before adding live/dead stain (1:100; Biolegend 423113), CD9-APC (1:100; Biolegend 312108) or CXCR4-APC (1:50; Biolegend 306510) for 30 min at 4 °C in the dark. Samples were washed 3× in FACS buffer. 100,000 events per sample were collected on BD LSRFortessa with FACSDiva 9.0 and analyzed in FlowJo 10.7.1. Gates were drawn on fluorescence minus one (FMO) controls.

**Immunohistochemistry**. Half brains were drop fixed in 4% (w/v) PFA for 48 h. The brains were cryoprotected in a 30% sucrose. Brains were sectioned coronally into 40 µm-thick slices on a freezing microtome (Leica SM 2010R) and stored in a solution of 0.05% NaN3 in 1× PBS as free-floating slices. For staining, tissue was blocked for 1 h in 1× PBS, 0.2% Triton X-100, and 10% goat serum. Immediately following blocking, brain sections were placed in primary antibodies diluted in 1× PBS and 1% goat serum and incubated overnight on a shaker at 4 °C. Samples were then incubated in conjugated secondary antibodies for 1 h followed by mounting on microscope slides. Sections were labeled with combinations of Amylo-Glo RTD Amyloid Plaque Stain Reagent (1:100; Biosensis TR-300-AG) (incubation for 20 min was needed before the addition of the primary antibodies), anti-GFP (1:500; Millipore Ab16901), anti-TagRFP (1:10,000; Kerafast EMU113), anti-Ku80 (1:250; Abcam ab79220), mounted with Fluoromount-G (SouthernBiotech). Additional samples were stained in anti-CXCR4 (1:100; MAB172 Clone 44716, R&D Systems), anti-CD9 (1:200; 312102, Biolegend), or anti-HLA-DRB1 (1:200; 14-9956-82, Invitrogen).

**Early postnatal intracerebroventricular transplantations**. Postnatal day 2–3 5xfAD-MITRG or WT-MITRG mice were placed in a clean cage over a heating pad with a nestlet from the home cage to maintain the mother's scent. Pups were then placed on ice for 2–3 min to induce hypothermic anesthesia. Free-hand transplantation was performed using a 30-gauge needle affixed to a 10 µL Hamilton syringe, mice received 1 µL of iHPCs suspended in sterile 1× DPBS at 62.5 total cells/µL at each injection site (see Hasselmann et al.[26]). Equal numbers of pups received mixed lines of GFP WT and RFP TREM2 KO HPCs (or the opposite combination). HPCs were thawed and plated into complete microglia medium and allowed to recover for 2 days prior to transplantation. Bilateral injections were performed at 2/5th of the distance from the lambda suture to each eye, injecting into the lateral ventricles at 3 mm and into the overlying anterior cortex at 1 mm, and into the posterior cortex in line with the forebrain injection sites, and perpendicular to lambda at a 45° angle. Transplanted pups were then returned to their home cages and weaned at P21. All mice were then sacrificed at 6 months of age as detailed below.

**Tissue dissociation for scRNA-seq and flow cytometry**. Following perfusion with ice-cold 1× DPBS containing 5 µg/ml actinomycin D, half brains were dissected, and the cerebellum was removed. Tissue was briefly stored on ice in RPMI 1640 containing 5 µg/mL actinomycin D, 10 µM triptolide, and 27.1 µg/mL anisomycin until subsequent perfusions were completed. Tissue dissociation was then performed utilizing the Tumor Dissociation kit, human (Miltenyi) and the gentleMACS Octo Dissociator with Heaters (Miltenyi) according to manufacturer guidelines with modifications. Briefly, tissue was cut into ~1 mm$^3$ pieces and placed into the C-tubes with the kit's enzymes, 5 µg/mL actinomycin D, 10 µM triptolide, and 27.1 µg/mL anisomycin and samples were dissociated using the pre-programmed protocol. Following enzymatic digestion, samples were strained through a 70 µm filter and pelleted by centrifugation. Myelin and debris were removed by resuspending the pellet in 8 mL 23% Percoll, overlaid with 2 mL of 1× DPBS, spinning at 400 × g for 25 min at 4 °C, with acceleration and brake set to 0, and discarding the myelin band and supernatant.

**Magnetic Isolation and FACS sorting of WT and TREM2 KO ex vivo microglia**. Dissociated cell pellets were resuspended in 160 µL FACS buffer (0.5% BSA in 1X DPBS) + 40 µL Mouse cell removal beads (Miltenyi) and incubated at 4 °C for 15 min. Magnetically labeled cells were then separated using LS columns and the MidiMACS separator (Miltenyi) while the unlabeled human cells were collected in the flow through. Isolated human cells were pelleted via centrifugation and prepared for cell sorting by resuspending in 400 µL of FACS buffer containing Hoescht 33342 (1:400) as a viability marker. For separation of human microglia expressing either endogenous GFP or RFP, samples were sorted on a FACS ARIA Fusion II (BD Biosciences) using the 70 um nozzle at the lowest flow rate. After removing doublets, by both forward and side scatter parameters, and selecting for live cells

(Hoescht 33342⁻), both GFP (GFP⁺RFP⁻) and RFP (GFP⁻RFP⁺) cells were gated on and sorted into individual tubes. For each cell type, 15,000 cells were sorted into collection tubes containing 5uL of FACS buffer, resulting in a final volume of ~20 µL.

**Single-cell sequencing**. scRNA-seq library preparation was performed according to the 10× Chromium Single Cell 3' Reagents kit v3 user guide and loading the full sample volume onto the Chromium chip (20 uL containing ~15,000 cells). The workflow was then followed according to the 10× protocol and samples were pooled at equimolar concentrations for sequencing on an Illumina HiSeq 4000. FASTQ files were aligned to both the human GRCh38.p12 transcriptome (Ensembl release 95) and the mouse mm10 transcriptome (Ensembl release 95)[70] using the CellRanger V3 count command, with the expected cells set to 5,000 and no secondary analysis performed. Following alignment, all reads that aligned to the mouse transcriptome were removed from the dataset before additional processing.

**Quantification and statistical analysis**

*In vitro experiments*. Data are presented as SEM and *n* represents the number of technical replicates of cell culture experiments. Unless specified, all experiments were carried out with all three isogenic knockout sets. Most statistical comparisons were conducted by ANOVA for Bonferroni's post hoc test in GraphPad Prism 7. For all experiments excluding RNA-sequencing and phagocytosis a statistically significant difference was defined as $p < 0.05$. For phagocytosis experiments, $p < 0.001$ was used.

*Image processing and plaque quantification*. Immunofluorescent sections were visualized and captured using an Olympus FV3000RS confocal microscope. Images represent confocal Z-stack taken with identical laser and detection settings. For human microglial proximity to plaque quantification, Z-stack images were taken at ×40 magnification (10 slices taken with a Z thickness of 1 µm), 4 images per hemisphere ($n = 3$ GFP WT & RFP TREM2 KO, $n = 3$ GFP TREM2 KO & RFP WT). Human microglia numbers and locations were detected and quantified through Ku80 (GFP/RFP positive) immunofluorescence using the Olympus cell-Sense imaging software. Cells located within the 50 µm radius from the center of the closest Amylo-Glo positive aggregate were detected. The distance of the center of the detected cells from the closest edge of the aggregate was then measured. Expression of HLA-DR or CD9 in GFP versus RFP cells in the same region were also quantified in Olympus CellSens 2.3. T-test run on Graphpad Prism 7.

*Bulk RNA-sequencing analysis*. RNA-sequencing data were processed and interpreted using either the Genialis visual informatics platform or in-house bioinformatic analysis pipelines. RNA-sequencing read integrity was verified using FastQC. BBDuk was used to trim adapters and filter out poor quality reads. Remaining reads were then mapped to the hg19 reference genome using the HISAT aligner or the GRCh38 reference genome (Ensembl release 97)[70] using Kallisto v0.46.0[71]. Complete clustering linkage was measured by pearson coefficient. Lowly expressed genes (expression count summed over all samples <10) were removed before differential expression analysis. Differentially expressed genes were calculated using DESeq2[72] by applying FDR cutoffs between 0.01 and 0.001 and log fold change of at least 0.5. Geneset enrichment analysis was performed using EnrichR GO Biological Processes 2018. Volcano plots were generated using the 'ggplot2' R package[73] and heatmaps were generated using the 'pheatmap' R package[74].

*Single cell RNA data visualization and differential gene analysis*. UMI count tables were read into Seurat (v3)[75] for preprocessing and clustering analysis. Initial QC was performed by log normalizing and scaling (default settings) each dataset followed by PCA performed using all genes in the dataset. Seurat's ElbowPlot function was used to select principal components (PCs) to be used for clustering along with a resolution parameter of 0.5 and clusters identified as being doublets, gene poor, or dividing were removed from the dataset prior to downstream analysis. Secondary QC cutoffs were then applied to retain only cells with less than 30% ribosomal genes, 15% mitochondrial genes, greater than 600 genes but less than double the median gene count, and greater than 500 UMI but less than double the median UMI count.

Cells passing QC were then merged according to mouse of origin (e.g., TREM2 WT and KO microglia from the same 5x-MITRG mouse) using Seurat's 'merge' function and data were processed using Seurat's wrapper for the 'sctransform' R package[76], while regressing out library size differences, percent ribosomal genes, and percent mitochondrial genes. PCA was then performed using the top 1000 variable genes after removing ribosomal and mitochondrial genes from the lists (WT: 930 genes; 5X: 968 genes). For xMGs isolated from MITRG mice, a shared nearest neighbor (SNN) plot was generated using Seurat's 'FindNeighbors' function using PCs 1:20 as input, clustering was performed using the 'FindClusters' function and a resolution parameter of 0.5, and dimension reduction was performed using the 'RunUMAP' function with the same PCs used for generating the SNN plot. xMGs isolated from the 5X-MITRG mouse were processed similarly, using PCs 1:14 and a resolution parameter of 0.3. Differentially expressed genes were determined between clusters using the 'FindAllMarkers' function, which employs a

Wilcoxon Rank Sum Test, with and FDR cutoff of 0.01, an LFC cutoff of 0.25, and the requirement that the gene be expressed in at least 10% of the cluster.

Information regarding the QC cutoffs and clustering parameters used in this analysis can be found in Supplementary Data 4.

**Reporting Summary**. Further information on research design is available in the Nature Research Reporting Summary linked to this article.

## Data availability

The datasets generated and analyzed in this study are available through GEO (bulk sequencing: GSE157652, GSE158469; single-cell sequencing: GSE158234). Any additional data presented in this paper is available from the corresponding author upon request. Source data are provided with this paper.

## Code availability

Base codes are open-source following citations linked in the methods section. Standard parameters were used unless otherwise noted in the methods section.

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

## Acknowledgements

This work was supported by NINDS T32 NS082174 (A.M), NIA T32 AG00096 (J.H.), NIH AG061895 (H.D.), AG016573 (J.S.S.), NIH NIA 5F30AG060704-02 and T32 AG000096 (G. H.), NRF 2020R1A2C2010285, 2020M3C7A1023941, and NIH AG059236-01A1 (H.C.), and NIH AG048099, NIH AG056303 and CIRM RT3-07893 (M.B.J.), Cure Alzheimer's Fund (Y.K. and H.C.) and a generous gift from the Susan Scott Foundation and HD Care. iPSC lines were generated by the UCI-ADRC iPS cell core funded by NIH AG016573. Experiments using the GFP-expressing iPSC line AICS-0036 and the RFP-alpha-tubulin iPSC line AICS-0031-035 iPSC were made possible through the Allen Cell Collection, available from Coriell Institute for Medical Research. The UCI Genomics High Throughput

Facility Shared Resource was supported by NIH grants P30CA-062203, 1S10RR025496-01, 1S10OD010794-01, and 1S10OD021718-01. We thank Christine Richardson at UNC Charlotte for helping FACS sorting of AD hNPCs and control hNPCs. Additionally, we would like to thank Cisbio for sharing their HTRF kits for TREM2. Finally, we would like to thank Dr. Samuel Marsh and the lab of Dr. Beth Stevens for providing expert guidance and cell isolation protocols for single-cell sequencing.

## Author contributions

M.B.J. and A.M. designed experiments. M.B.J., A.M., and A.J. prepared the paper. C.H.T. and J.P.C. generated iPSC cells and CRISPR isogenic lines. E.M. A.S., and I.S. performed stimulation for RNA sequencing and initial studies. A.M. performed survival and phagocytosis assays. G.F., L.M.T., and J.S.S. prepared the APOE. Y.J.K., A.M., and H.C. designed and performed in vitro chemotaxis assay. A.J., A.M., and M.Ca. designed and performed calcium imaging. J.Si, C.C., and G.A.P. provided training and isolated synaptosome fragments. M.Co. and H.D. performed in vivo studies. M.Co., S.K.S., A.S., E.D., and M.B.J. analyzed in vivo studies. A.M. and J.H. analyzed sequencing data. All authors read and approved the final paper.

## Competing interests

M.B.J. is a co-inventor of patent WO/2018/160496 related to differentiation of human pluripotent stem cells into microglia. All other authors declare no competing interests.
