## [Peer Review File · Nature Communications]

Reviewers' Comments:

Reviewer #1:

Remarks to the Author:

McQuade and colleagues study the effect of TREM2 deficiency on human iPS-derived microglia, both in vitro and in a chimeric amyloid mouse model.

The authors present an elegant set of data to address the effect of loss of TREM2 in human microglia. Overall the data are novel and intriguing and of interest to the field of microglia/AD and approached in a highly sophisticated mouse.

Nonetheless, there are several points that need attention and to be addressed:

1) The authors state that the mouse model presented here is described in reference 27, Hasselmann, Jonathan et al. Development of a chimeric model to study and manipulate human microglia in vivo. However, at the time of this reviewing, this paper was not online, not on the Neuron website, not on PubMed. So it is impossible to judge what was shown in that paper, how it validates the system, if there is overlap with the data presented here etc.

2) For RNA-seq, n=3 is used. It is unclear if this is 3 differentiations from 1 iPS line, or 1 differentiation for 3 iPS lines, or possibly a mix of these options. Was this done in a single iPS \diamond microglia differentiation experiment, or multiple? How reproducible and robust are these findings, gene expression changes. This needs to be clarified and substantiated.

3) How was the heatmap in 1F generated. Is it supervised/unsupervised clustering of these 72 genes. The colors do not really facilitate interpretation. The green and blue are quite similar, especially in print.

4) Interpretation of the sc-Seq data would be facilitated if bar plots were provided where it is indicated what the cluster contribution of each respective genotype is.

5) Why were the sc-Seq data from MITGR and 5x-MITRG microglia analyzed separately. To really show shifts in homeostatic genes/subclusters it is required to not only analyze the MITGR and 5x-MITRG microglia separately but also together.

6) How does the expression of typical homeostatic/DAM/HLA/apoptotic genes look like when plotted in the tSNEs or plotted in violin plots look like?

7) How many cells were put in the Chromium, recovered per genotype and per treatment and how many genes were on average detected/cell in the sc-Seq data. From how many mice were the sc-Seq data generated. 1 mouse or multiple mice/group? Were the samples from 1 treatment group pooled? Overall, more info/metrics on the sc-Seq study design needs to be provided to facilitate interpretation of the sc-Seq data.

8) Were the mouse microglia in these mice also analyzed in parallel to the human iPS-derived microglia to substantiate the claim that the human microglia do not acquire a DAM state where mouse microglia do in this AD mouse model? Again, ref 27 was not accessible to this reviewer.

9) how do the endogenous mouse microglia respond/react to the human iPS derived cells. Is there a difference between the WT and TREM-KO hiPSC-microglia? Does that affect amyloid aggregation and association of hiPS-microglia with these aggregates?

Reviewer #2:

None

Reviewer #3:

Remarks to the Author:

While TREM2 mouse models have provided critical insight, the normal and disease-associated functions of TREM2 variants in human microglia remain unclear. To examine this question, the authors developed 3 isogenic, CRISPR-modified TREM2 knockout induced pluripotent stem cell lines. Microglia derived from these lines were transplanted into mice to generate a novel chimeric model. Using transcriptomic and functional analyses they report that TREM2 deletion produces robust changes in the migration, survival, and phagocytic capacity of human microglia, and an impaired response to beta-amyloid plaques in vivo. Single-cell sequencing demonstrates a loss of disease-associated microglial (DAM) responses in TREM2 knockout human microglia and the appearance of a new apoptotic cluster in diseased animals. There are several novel features of this manuscript that are of general interest and importance to the AD field and neurodegenerative disease/aging generally. The development of a novel chimeric model in which crispr genome editing is used to generate isogenic mutant lines that can be introduced to a common mouse genetic background is potentially important and interesting. However, much of the data confirms previous findings in WT and TREM2 KO mouse microglia. While the concept of the paper is interesting several key pieces of information are not well described or the robustness of the analyses and/or observations is questionable as outlined below.

MAJOR:

1. In general, this study confirms published findings of WT and TREM2 KO mouse microglia in control and 5xFAD mice. However, the authors report one major novel finding that human microglia behave differently than mouse microglia in respect to TREM2 complete loss-of-function in the appearance of what they refer to as an "apoptotic" cluster. However, in order to conclude this (that species of origin is the causal factor in the appearance of this cluster), the authors need to exclude the possibility that transplantation, mouse genetic background and/or clustering methods produced this result. Therefore, in order to make such claim, the authors should transplant WT and TREM2 KO mouse microglial cells in the MITRG and 5x-MITRG mice and perform the same analysis as they did with WT and TREM2 KO human microglial cells, and compare/contrast the two analyses using, e.g., LIGER (<https://macoskolab.github.io/liger/>).

2. In the section describing the in vivo experiments related to the DAM response, the authors should justify and explain why a different transplantation model has been used here compared to the previous in vivo experiments in the 5x-MITRG mice, for which a failed response to amyloid deposition was already demonstrated. Also, how many independent clones were derived from the GFP and RFP expressing lines and tested in the experiments that follow? And how do they relate to the clones used in previous in vivo experiments in the 5x-MITRG mice?

In Fig, 4A, the authors report finding 4 distinct clusters/sub-populations of WT hMGL in MITRG mice. Given that Seurat performs clustering based on a user-defined resolution parameter (and dimensionality reduction, which affect clustering, based on the perplexity parameter) to determine the number of clusters, the authors should justify the use of their chosen perplexity and resolution parameters and robustness of the 4 clusters based on parameter sensitivity analysis. Since Figure 4A clearly demonstrates that the 4 clusters are not readily separated and are not clearly distinct (although the t-sne distance does not necessarily reflect differences in transcriptomes), the authors should assess the stability of this clustering, using iterative statistical frameworks, such as iterclust (Ding et al. 2018). To validate the user-driven clustering obtained through Seurat, the authors should use an alternative data-driven and unsupervised clustering algorithm, such as PhenoGraph (Levine et al. 2015). Along the same lines, in the legend for Figure 4A, the authors mention that "Within 5x-MITRG mice, WT and TREM2 KO cells show far greater separation in tSNE clustering". T-SNE is not a clustering algorithm, but merely a dimensionality reduction technique (just more advanced than PCA); additionally, the distance between clusters usually is not informative in t-SNE plots and should not be interpreted so readily. Additionally, since the authors cluster the cells from MITRG and 5x-MITRG mice separately, this comparison is also not applicable.

Also, the assignment of biological function to the 4 clusters should be more clearly explained and/or executed beyond visual inspection of top genes. What percentile was used, what significance threshold, was pathway analysis used to assign function(s) based on the DEGs? The same applies to the 6 clusters of WT hMGL in the 5x-MITRG mice. For example, the authors identify a homeostatic population in Figures 4A and 4B; yet lower t-sne plots demonstrate that the markers for homeostatic microglia are highly expressed in all populations. This observation seems to indicate the populations are not robustly identified. Same applies to the apoptotic population that does not show expression of apoptotic markers on the lower t-sne plot. These discrepancies should be addressed.

When the authors mention that the cell population observed is "highly similar" to DAM, this overlap should be tested statistically.

When comparing WT and TREM2 KO hMGL cells in MITRG and 5x-MITRG mice, the authors should cluster the cells from all experiments together. There are several methods for "meta-analyzing" multiple experiments (Welch et al. 2019; Stuart et al. 2019).

The claim that the apoptotic microglia population is a novel population only observed in 5x-MITRG TREM2 knockout microglia can only be made if the datasets are analyzed together, cell proportion differences are properly tested and the cluster is robust and pathway analysis corroborates this claim (the authors mention that this cluster is very small). This applies to all other claims of "shifts in microglial populations"; the authors need to establish or use an existing statistical framework (Liu et al. 2018; Farbehi et al. 2019) to formally test how significant these shifts are and how much these changes could be attributable to chance given the variability across samples.

In Fig. 4C, it is unclear why TREM2 KO microglia are identified by absence of GFP fluorescence when RFP fluorescence is available to more directly identify them. Also, statistical analysis of the imaging findings should be reported (e.g., number of microglia clustered around plaques, changes in homeostatic, HLA and DAM marker intensity, etc.). Also, the presence of apoptotic cells in TREM2 KO vs WT hMGL should be validated using TUNEL, Caspase-3/7 or similar assays.

3. The use of clonal cell lines requires statistical analysis of multiple independent clones. Importantly, the authors generated "three isogenic sets of WT and TREM2 KO hiPSCs" (line 1-3 in Fig. 1A). Are these WT and TREM2 KO clones from three independent derivations, three different individuals, or what? More details about donor and derivations are needed. Also, in the Methods section, the authors explain that n refers to technical replicates for each experiment and that biological replicates are considered to be independent experiments performed at different dates, using - unless specified, but it is never specified - all three lines mentioned above. Since the observational unit (and thus the reported n) in these statistical analyses should be, for the experiments described here, the line and not the technical replicates, more details about how the technical and "biological" replicates as defined by the authors were collapsed at the level of the observational unit, the line (e.g., by averaging measurements across technical replicates at different dates for each line, and then applying the statistical test for $n = 3$ where n is the number of lines, or using a hierarchical model?). More importantly, this detailed information about experimental design and statistical analysis should be extended to the in vivo experiments where a clear indication of how many lines were used is necessary. As an example, the heatmap in Fig. 1F shows 4 samples per combination of genetic background and stimulation: how many of these 4 samples are technical vs "biological" replicates? For each sample, the line used should be reported. In Fig. 1H, n is not reported.

4. Suppl. Table 1/2 format is very confusing. Please provide two separate sheets, one including information about all genes analyzed (not just the DEGs) along with gene ID (hugo, ensembl or entrez id), gene symbol, LFC and adjusted pvalue; and the other sheet including information about the significantly enriched genesets (GO, Kegg, etc), along with the pvalue of enrichment adjusted for the total number of terms/pathways tested (total number should be reported), total number of genes in the geneset as well as gene symbols and IDs of the DEGs that drives the enrichment. The adjusted pvalue threshold used for the genesets should be reported. The source and version/date

of the genesets should also be reported. The authors should also refer to these analyses as geneset enrichment analyses and not GO enrichment analyses, since Kegg and other genesets were used in addition to GO genesets.

5. For the experiments illustrated in Fig. 1, the authors used MAb-anti-TREM2 to stimulate TREM2 signaling as opposed to the TREM2 KO condition the authors used to obtain TREM2 loss-of-function. We argue that a better contrast to the TREM2 KO condition is over-expression of TREM2 in order to obtain gain-of-function. Indeed, the authors point out that MAb treatment may induce rapid internalization of TREM2 thus leading to a loss-of-function, particularly when considering the extremely transient activation of signaling downstream of TREM2 (minutes) compared to when the assays were performed (1 day post-treatment). Also, MAb-anti-TREM2 is not a disease-relevant stimulus (the authors used synaptosomes and fibrillar Abeta as disease-relevant stimuli in subsequent experiments). For these reasons, the transcriptome profiling would be a lot more informative if repeated using TREM2 over-expression and, for examples, synaptosomes, fibrillar Abeta or, even better, apoptotic cells as the disease-relevant stimulus, given that R47H mutation impairs binding of TREM2 to PtdSer and apoptotic cells. For the same reason, analysis of efferocytosis in Fig. 2 would be highly desirable. Also, how do the authors reconcile their findings that TREM2 loss-of-function (which is AD risk-increasing) leads to impaired synaptosome phagocytosis (which the authors claim to be a proxy for synaptic pruning) when increased synaptic pruning has been shown in mouse models to worsen disease-related phenotypes and proposed to be bad for AD, e.g., by Beth Stevens.

6. The authors keep referring to previous studies claimed to show decreased microglial migration in TREM2 KO compared to WT mice. However, these studies have only shown decreased number and altered morphology/activation state of microglial cells around plaques. This phenotype could be due to altered migration but also altered survival and/or proliferation, thus it should not be assumed, until directly observed or otherwise demonstrated, that this defect is the result of altered migration. Because the authors have also not demonstrated migratory deficits in their in vivo experiments, they should not use the words "migratory deficits/migration" when describing their results in Fig. 3A. In Fig. 3A, the authors should specify what observational unit was used to derive the statistics shown in the bar plot (plaque, field, mouse? How many plaques/fields were used per mouse? Were the analyses blinded?). Related to this, and as previously requested, also the breakdown of the n into biological and technical and line replicates is needed for the reader to understand how many independent clonal cell lines were used in this experiment and whether the effect was consistent across the different clones. Also, when printed, the tiny colored spots are really difficult to see on the black background. Perhaps using false coloring may help the reader visualize the position of microglia in relation to the other features (plaques, circles, etc).

In Fig. 3A, plaque morphology (and perhaps number) appears to be different between WT and TREM2 KO. Are these differences significant? Are these fields taken from matched regions in the brain? How do the authors explain such difference?

In Fig 3B, how do these concentrations and ratios of Abeta40 and 42 compared to what's measured in human AD brains? How do the authors explain such difference in response?

In Fig 3C, is the migration toward AD neurons vs healthy neurons due to apoptosis in the AD neurons cultured for 9w vs 3w? Since TREM2 has been shown to mediate the response of microglia to apoptotic cells, please repeat the experiment using live vs apoptotic cells (Jurkat cells for example) to ascertain whether the migratory response is due to Abeta or cues present on apoptotic cells in general. Also the title of Fig. 3 is misleading because microglia don't appear to respond robustly to healthy neurons and lack of TREM2 has not been tested in these experiments to ascertain whether it affects migration toward healthy neurons. This additional condition should be added to these experiments. Moreover, what are the levels of soluble Abeta 40 and 42 in the conditioned medium at the two different time points for the healthy and AD neurons?

In Fig. 3D and E, if the authors want to make a more solid claim about the role of CXCR4 in the migratory deficit observed in Fig. 3C, they should deplete CXCL12 using an antibody and show the effect in the assays used in Fig. 3C.

MINOR:

“Microglia perform key macrophage-like functions” -> Microglia are macrophages and perform the same core functions of all macrophages (e.g., efferocytosis and immune surveillance/response).

“With the identification of several immune genes that alter risk for AD” -> AD GWAS studies identify loci, not genes. With a few exceptions, including TREM2, we don't have, at the moment, additional evidence to identify or prioritize causal genes in AD loci. Also, TREM2 is not microglia-specific and is expressed on several other tissue-resident macrophages, including non-parenchymal brain macrophages. A role for TREM2 function in this other myeloid cell populations in the modulation of AD risk cannot be excluded.

Main text refers to Suppl. Table 1 to include 444 DEGs yet the table contains 695 DEGs, this discrepancy should be explained and corrected.

Main text refers to Suppl. Table 2 to include 114 DEGs yet the table contains 335 DEGs, this discrepancy should be explained and corrected.

In Fig. 1F, why is gene expression reported as negative log of TPM+1 and why does this measure shows positive values? Normally expression in RNA-Seq experiments is reported as $\log(\text{TPM}+1)$ which is a positive number starting from zero for non-expressed genes.

In Fig. 1H one of the treatments is called 'no medium'. This is confusing. It should say -IL34,-MCSF,-TGFB1 or "no medium change". In an ideal experimental design, medium should have been changed with the same medium lacking the three growth factors to exclude medium changing as a factor in the experiment.

In Fig. 1, the graphical figure legends are confusing, with colors and gray scale trying to convey lighter and darker colors. The different colors should all be clearly explained or another graphical attribute (e.g., line pattern or shape) used to visualize WT vs TREM2 KO.

In Fig. 2 the authors mention Zymosan as specific to Dectin-2 when it has been shown to be a ligand for Dectin-1 as well. Also both Dectins signal through Syk phosphorylation... how do the authors explain the discrepancy between TREM2 ligands and zymosan phagocytosis when inhibiting Syk phosphorylation?

We would like to thank the reviewers for their supportive, insightful, and highly constructive comments. We have now performed several additional experiments, re-analyzed the RNA sequencing data per their suggestions, and provided additional methodological details. We hope you will agree that the manuscript is further improved and provides important new insight into the function of TREM2 in human microglia and Alzheimer's disease pathogenesis. Below are point by point responses to each of the Reviewer's comments.

Reviewer #1 (Remarks to the Author):

McQuade and colleagues study the effect of TREM2 deficiency on human iPS-derived microglia, both in vitro and in a chimeric amyloid mouse model.

The authors present an elegant set of data to address the effect of loss of TREM2 in human microglia. Overall the data are novel and intriguing and of interest to the field of microglia/AD and approached in a highly sophisticated mouse.

Nonetheless, there are several points that need attention and to be addressed:

1) The authors state that the mouse model presented here is described in reference 27, Hasselmann, Jonathan et al. Development of a chimeric model to study and manipulate human microglia in vivo. However, at the time of this reviewing, this paper was not online, not on the Neuron website, not on PubMed. So it is impossible to judge what was shown in that paper, how it validates the system, if there is overlap with the data presented here etc.

We apologize that this paper was not provided to you at the time of review. It was uploaded with our submission as a supporting document, but apparently was not forwarded to the reviewers. We fully agree that this other study was critical to understanding the chimeric experiments conducted in the current manuscript. This reference has now been published in the journal *Neuron* and is available via the following link (<https://www.ncbi.nlm.nih.gov/pubmed/31375314>).

2) For RNA-seq, n=3 is used. It is unclear if this is 3 differentiations from 1 iPS line, or 1 differentiation for 3 iPS lines, or possibly a mix of these options. Was this done in a single iPS \diamond microglia differentiation experiment, or multiple? How reproducible and robust are these findings, gene expression changes. This needs to be clarified and substantiated.

We had originally completed the RNA-sequencing with 4 replicates from the same iPS background since we validated the major conclusions from that data with functional analyses. However, we have now repeated these RNA sequencing experiments using two isogenic pairs of WT and TREM2 KO iPSCs generated from two different individual human subjects. For each of these pairs of lines we also used n=3 technical replicates for each treatment. We have also more clearly stated these important details within the manuscript. "Two independent sets of TREM2 isogenic microglia generated from two distinct patients were sequenced to account for any cell line-dependent effects"

3) How was the heatmap in 1F generated. Is it supervised/unsupervised clustering of these 72 genes. The colors do not really facilitate interpretation. The green and blue are quite similar, especially in print.

The heatmap from 1F (now 1J) was generated through unsupervised clustering of the 72 significantly altered genes. Both axes are unsupervised. For the colors, we used the Viridis color pallet (<https://cran.r-project.org/web/packages/viridis/vignettes/intro-to-viridis.html>) which is specifically developed for use in black and white as well as being optimal for understanding with people in various types of color blindness. This pallet has been empirically shown to be more quickly recognizable than monochromatic pallets (http://delivery.acm.org/10.1145/3180000/3174172/paper598.pdf?ip=169.234.85.197&id=3174172&acc=ACTIVE%20SERVICE&key=CA367851C7E3CE77%2EE385B6E260950907%2E4D4702B0C3E38B35%2E4D4702B0C3E38B35&acm_=1567810075_281cf7f66b36fb64854cdcd340f)

86f03).

4) Interpretation of the sc-Seq data would be facilitated if bar plots were provided where it is indicated what the cluster contribution of each respective genotype is.

Thank you for this excellent suggestion. We have now added bar graphs to help with interpretation of the data. In addition, we added a supplemental table showing the fold change of each cluster between WT microglia and TREM2 knockout microglia (Supplementary Table 4).

5) Why were the sc-Seq data from MITGR and 5x-MITRG microglia analyzed separately. To really show shifts in homeostatic genes/subclusters it is required to not only analyze the MITGR and 5x-MITRG microglia separately but also together.

Due to real biological variance between the AD-model and non-diseased mice, we have found that the Seurat scSeq analysis pipeline does not merge the WT and 5XfAD samples together effectively. Using the Integrated Analysis pipeline from Seurat V3 appears to overcorrect the data and artificially assigns 2 % WT cells to the DAM cluster. This was concerning to us as we are unable to detect and validate the existence of "DAM-like" cells in the WT animal histology by protein expression of several DAM markers (HLA-DRB1, CD9, LGALS3) that strongly label plaque-associated microglia in the 5x-MITRG mice. Existing single cell sequencing pipelines were primarily developed to distinguish relatively divergent cell types (ie: neurons vs. microglia). When these pipelines are applied to only a single cell type, they have a tendency to overcorrect the data when biologically different groups are clustered together. As this combined analysis does not correlate well with our IHC data, we present the separate analysis of 5x-MITRG and MITRG mice in Figure 6. However, we have now also included a combined analysis of this data in Supplemental Table 6.

6) How does the expression of typical homeostatic/DAM/HLA/apoptotic genes look like when plotted in the tSNEs or plotted in violin plots look like?

The plots in the previously submitted manuscript showed expression of typical gene families together. To address this question, we have now also included specific gene UMAP graphs within Supplemental Figure 5. In addition we have included heatmaps to show expression of these genes across all clusters in the same supplemental figure.

7) How many cells were put in the Chromium, recovered per genotype and per treatment and how many genes were on average detected/cell in the sc-Seq data. From how many mice were the sc-Seq data generated. 1 mouse or multiple mice/group? Were the samples from 1 treatment group pooled? Overall, more info/metrics on the sc-Seq study design needs to be provided to facilitate interpretation of the sc-Seq data.

We appreciate this important point. The methods have been revised to specify that the full sample volume onto the Chromium chip (20 uL containing ~15,000 cells). Additionally, we have included a supplemental table (Supplemental Table 5) that contains the information pertaining to the total cells captured from each individual sample, cells remaining after quality control, and the average and median UMI and gene counts at each step. We have also clarified how the samples were bioinformatically combined within the methods section.

8) Were the mouse microglia in these mice also analyzed in parallel to the human iPS-derived microglia to substantiate the claim that the human microglia do not acquire a DAM state where mouse microglia do in this AD mouse model? Again, ref 27 was not accessible to this reviewer.

Unfortunately, mouse microglia were not analyzed in parallel to the human samples from this model. We had transplanted additional mice to try to address this critique, but due to the COVID pandemic and the resulting shut down of research and core activities, we were unable to perform additional single cell sequencing. It's is however worth noting that we specifically chose to cross

the MITRG with the 5xFAD model because of the existing murine data from Keren-Shaul et al. in this model. In our recently published manuscript (Hasselmann et al. *Neuron* 2019) we compare mouse and human responses to AD pathology and find a relatively limited overlap between mouse and human transcriptomic responses which we also discuss in this manuscript.

9) How do the endogenous mouse microglia respond/react to the human iPS derived cells. Is there a difference between the WT and TREM2-KO hiPSC-microglia? Does that affect amyloid aggregation and association of hiPSC-microglia with these aggregates?

Our recent publication on the development of this chimeric model demonstrates that within the forebrain 80% of all microglia are human. In addition, our transplantation design utilized co-transplantation of TREM2 WT and KO microglia (distinguishable via GFP and RFP expression). As a result, we consistently find that the WT human microglia are associated with plaques and express DAM markers whereas the TREM2 KO cells show diminished plaque association and an absence of DAM markers. Because of this co-transplantation approach an assessment of the impact of TREM2 genotype on amyloid aggregation cannot be examined with high fidelity. Future models that genetically lack murine microglia will hopefully provide a more clear answer to this question. However, these mice are still being established and will require over a year before they can be validated.

Reviewer #3 (Remarks to the Author):

While TREM2 mouse models have provided critical insight, the normal and disease-associated functions of TREM2 variants in human microglia remain unclear. To examine this question, the authors developed 3 isogenic, CRISPR-modified TREM2 knockout induced pluripotent stem cell lines. Microglia derived from these lines were transplanted into mice to generate a novel chimeric model. Using transcriptomic and functional analyses they report that TREM2 deletion produces robust changes in the migration, survival, and phagocytic capacity of human microglia, and an impaired response to beta-amyloid plaques in vivo. Single-cell sequencing demonstrates a loss of disease-associated microglial (DAM) responses in TREM2 knockout human microglia and the appearance of a new apoptotic cluster in diseased animals. There are several novel features of this manuscript that are of general interest and importance to the AD field and neurodegenerative disease/aging generally. The development of a novel chimeric model in which CRISPR genome editing is used to generate isogenic mutant lines that can be introduced to a common mouse genetic background is potentially important and interesting. However, much of the data confirms previous findings in WT and TREM2 KO mouse microglia. While the concept of the paper is interesting several key pieces of information are not well described or the robustness of the analyses and/or observations is questionable as outlined below.

MAJOR:

1. In general, this study confirms published findings of WT and TREM2 KO mouse microglia in control and 5xFAD mice. However, the authors report one major novel finding that human microglia behave differently than mouse microglia in respect to TREM2 complete loss-of-function in the appearance of what they refer to as an “apoptotic” cluster. However, in order to conclude this (that species of origin is the causal factor in the appearance of this cluster), the authors need to exclude the possibility that transplantation, mouse genetic background and/or clustering methods produced this result. Therefore, in order to make such claim, the authors should transplant WT and TREM2 KO mouse microglial cells in the MITRG and 5x-MITRG mice and perform the same analysis as they did with WT and TREM2 KO human microglial cells, and compare/contrast the two analyses using, e.g., LIGER (<https://macoskolab.github.io/liger/>).

Thank you for this comment as it led us to further investigate this cluster before moving forward. Upon performing immunohistochemistry, we found that these apoptotic cells were not detectable in the brain and thus have re-analyzed the data to exclude this cluster as it may be an artifact of

the isolation protocol. It certainly has been shown with microglia that long isolation procedures can alter the transcriptome of these reactive cells. We have aimed to shorten our isolation protocol as much as possible and do include transcription inhibitors throughout the process, but since we do not see evidence of the apoptotic cluster in the brain, it is likely that this phenotype is instead induced through isolation. We do now include discussion of this information in the conclusion section as we feel it is important for the field to recognize these caveats in all microglia sequencing experiments. Additionally, this apoptotic cluster is of interest in that it does only occur in the TREM2 KO microglia from an AD brain. Even if these cells are not present in the brain, the fact that the same isolation procedure only produced apoptotic cells in one sensitive cell type/mouse genotype combination is still of interest and may indeed represent a true sensitivity of these cells to perform apoptosis more quickly in the face of stress. This data parallels our findings *in vitro* from Figure 2 which we wanted to highlight in discussion.

We have taken your comments to heart and re-formatted the paper to rely less heavily on the discovery of this cluster, but feel it is still important to discuss while cognizant of the specific circumstances that may have created it.

2. In the section describing the *in vivo* experiments related to the DAM response, the authors should justify and explain why a different transplantation model has been used here compared to the previous *in vivo* experiments in the 5x-MITRG mice, for which a failed response to amyloid deposition was already demonstrated. Also, how many independent clones were derived from the GFP and RFP expressing lines and tested in the experiments that follow? And how do they relate to the clones used in previous *in vivo* experiments in the 5x-MITRG mice?

In response to this comment and provide a more consistent approach throughout the manuscript we performed a new set of transplantations using the optimal postnatal transplantation paradigm. The data now provided in Figure 4A was collected from these new P2 transplants. Thus, all *in vivo* experiments have now been conducted using the same postnatal transplantation model. In all transplant experiments in this manuscript, we used two independent TREM2 knockout lines, one in the RFP-expressing line and another in the GFP-expressing line. Mice were transplanted with both combinations of cells, RFP WT/GFP TREM2 KO as well as GFP WT/RFP TREM2 KO. We suggest that given the novelty of the chimeric approach and the high cost of these *in vivo* experiments, using two different isogenic knockout sets is sufficient. These are the same lines used to analyze plaque proximity in Figure 4.

In Fig, 4A, the authors report finding 4 distinct clusters/sub-populations of WT hMGL in MITRG mice. Given that Seurat performs clustering based on a user-defined resolution parameter (and dimensionality reduction, which affect clustering, based on the perplexity parameter) to determine the number of clusters, the authors should justify the use of their chosen perplexity and resolution parameters and robustness of the 4 clusters based on parameter sensitivity analysis. Since Figure 4A clearly demonstrates that the 4 clusters are not readily separated and are not clearly distinct (although the t-sne distance does not necessarily reflect differences in transcriptomes), the authors should assess the stability of this clustering, using iterative statistical frameworks, such as iterclust (Ding et al. 2018). To validate the user-driven clustering obtained through Seurat, the authors should use an alternative data-driven and unsupervised clustering algorithm, such as PhenoGraph(Levine et al. 2015). Along the same lines, in the legend for Figure 4A, the authors mention that “Within 5x-MITRG mice, WT and TREM2 KO cells show far greater separation in tSNE clustering”. T-SNE is not a clustering algorithm, but merely a dimensionality reduction technique (just more advanced than PCA); additionally, the distance between clusters usually is not informative in t-SNE plots and should not be interpreted so readily. Additionally, since the authors cluster the cells from MITRG and 5x-MITRG mice separately, this comparison is also not applicable.

Thank you for this comment. We acknowledge that the clusters are not well separated, which appears to be a product of our cells being more similar than they are different, since we are analyzing a single cell type rather than the multiple cell types that most single-cell experiments examine. However, we have taken precautions to ensure that the chosen analysis parameters

are robust, and we have provided the reviewers with a figure (below) detailing the typical iterative approach being used to determine these parameters. We have also taken this reviewer's suggestion to compare our clustering to that of a data-driven, unsupervised approach such as Phenograph, and have found that the clustering is similar, although Seurat appears to be more capable of clearly distinguishing the identified clusters with the added benefit of being able to differentiate the DAMs from the HLA cluster. If the reviewer thinks it is important to add the below analysis as a supplemental figure, we would be happy to. Additionally, we acknowledge that our choice of terminology was poor and we have carefully edited the text to ensure accurate use of the terms relating to this analysis.

iterative clustering of scRNA data to determine optimal parameters. A) Seurat's ElbowPlot

command was used to generate a graphic which organized the principal components (PC) in order of decreasing explained variance. Multiple cutoffs were selected for subsequent iterative analysis. B) The PC cutoffs were used as input for the “dims” argument of Seurat’s FindNeighbors command in order to generate a shared nearest neighbor plot, values between 0.1 and 1.1 were used as input for the resolution parameter of the FindClusters command, and results were visualized using the UMAP dimension reduction technique. Representative results for PCs 1-14 are shown but the process was completed for each cutoff. C) Heatmaps, representing the clustering from B, show that as the resolution parameter is increased above 0.3, the cells begin to be grouped into smaller clusters that are distinguished by less prominent gene expression differences. D) Using Phenograph to cluster the data with a k parameter of 100, clusters were found that are high in MHCII gene expression, interferon (IFN) gene expression, and two clusters that are low in both MHCII and IFN gene expression. Notably, Phenograph was unable to distinguish the DAM cluster from the MHCII cluster. E) UMAP visualizations of both the Phenograph and Seurat clustering show that, while the core clusters are similar, Seurat could distinguish the DAM cluster and more clearly separated the MHCII and IFN clusters from the Homeostatic cluster.

Also, the assignment of biological function to the 4 clusters should be more clearly explained and/or executed beyond visual inspection of top genes. What percentile was used, what significance threshold, was pathway analysis used to assign function(s) based on the DEGs? The same applies to the 6 clusters of WT hMGL in the 5x-MITRG mice. For example, the authors identify a homeostatic population in Figures 4A and 4B; yet lower t-sne plots demonstrate that the markers for homeostatic microglia are highly expressed in all populations. This observation seems to indicate the populations are not robustly identified. Same applies to the apoptotic population that does not show expression of apoptotic markers on the lower t-sne plot. These discrepancies should be addressed. When the authors mention that the cell population observed is “highly similar” to DAM, this overlap should be tested statistically. When comparing WT and TREM2 KO hMGL cells in MITRG and 5x-MITRG mice, the authors should cluster the cells from all experiments together. There are several methods for “meta-analyzing” multiple experiments (Welch et al. 2019; Stuart et al. 2019).

Thank you for encouraging us to make this aspect of the paper more clear. We have now clarified our descriptions regarding the gene families used to describe the clusters in our single-cell data. However, as we have not experimentally determined the biological function of the clusters, we were not attempting to assign function to any specific cluster. Rather, we sought to describe the clusters based on the cellular markers that define them and, as we do not feel that scRNA-seq data is powerful enough to yield robust pathway analysis, we have chosen to manually curate the genes that are present in the clusters. As such, the cluster that is high in genes that are part of the Major Histocompatibility Complex II (MHCII) has been deemed HLA. Similarly, the IFN cluster is defined by genes that can be found in Gene Ontology pathways related to the Type 1 Interferon response and the DAM cluster has been defined based on our previous publication (Hasselmann et al. (2019)) as this was the first paper to describe the DAM phenotype in human microglia. Finally, our definition of the “Homeostatic” cluster despite a number of those genes also being expressed in the HLA and IFN clusters, is due to that fact that the “Homeostatic” cluster is also low in both the HLA and IFN markers while also being high in traditional microglia marker of homeostasis such as CX3CR1, P2RY12, and P2RY13.

The full DEG lists have also been made available as supplemental tables (Supplementary Table 5) so that other researchers can verify or dispute our manual curation of these clusters. A great deal of additional research that we believe is beyond the scope of the current manuscript will be needed to determine the true functional differences between these clusters of microglia.

The claim that the apoptotic microglia population is a novel population only observed in 5x-MITRG TREM2 knockout microglia can only be made if the datasets are analyzed together, cell

proportion differences are properly tested and the cluster is robust and pathway analysis corroborates this claim (the authors mention that this cluster is very small). This applies to all other claims of “shifts in microglial populations”; the authors need to establish or use an existing statistical framework (Liu et al. 2018; Farbehi et al. 2019) to formally test how significant these shifts are and how much these changes could be attributable to chance given the variability across samples.

In this resubmission, we have aimed to rely less heavily on the sequencing data and decided to use this data to better inform histological protein analysis *in vivo*. At the protein level, it did become very clear that indeed these shifts of microglia populations do occur between TREM2 WT and KO cells (see Figure 6c-e).

In response to your concerns, we did re-analyze the single-cell sequencing data together using Seurat CCA. While we did still uncover the 5x-specific cluster (DAMs) we feel that this analysis is over-clustering the data and is still as subjective as our previous analysis. However, we have included this analysis's clustering as a supplemental table (S6) if readers wish to see this clustering paradigm as well.

As mentioned above we were unable to confirm the existence of apoptotic cluster cells in the brains by histology and thus have removed this cluster from the sequencing data as we suspect that it may occur in response to isolation.

In Fig. 4C, it is unclear why TREM2 KO microglia are identified by absence of GFP fluorescence when RFP fluorescence is available to more directly identify them. Also, statistical analysis of the imaging findings should be reported (e.g., number of microglia clustered around plaques, changes in homeostatic, HLA and DAM marker intensity, etc.). Also, the presence of apoptotic cells in TREM2 KO vs WT hMGL should be validated using TUNEL, Caspase-3/7 or similar assays.

We had previously identified our microglia using only GFP+ Ku80+ and GFP- Ku80+ because the endogenous RFP fluorescence was quenched by fixation of the tissue. We have now validated an anti-RFP antibody to include in our staining protocol so that we may identify more directly GFP+ and RFP+ cells. Additionally, we have now quantified distance to plaques from this P2 model in the migration figure (Figure 4) and included an additional metric- now looking at both raw distance to plaques as well as the percent of each genotype found within 50 um of the plaque.

For intensity of cluster markers, we now show quantification of markers from the most important changed clusters (CD9 and MHCII). Our original intent was to merely use the images as validation of the sequencing data, but we agree that quantification of these experiments in a larger cohort of mice greatly strengthens our argument. Thank you for that suggestion! As the apoptotic cluster was not detected histologically it is no longer included in this figure.

3. The use of clonal cell lines requires statistical analysis of multiple independent clones. Importantly, the authors generated “three isogenic sets of WT and TREM2 KO hiPSCs” (line 1-3 in Fig. 1A). Are these WT and TREM2 KO clones from three independent derivations, three different individuals, or what? More details about donor and derivations are needed.

Thank you for raising this important point, we have now included this data in the methods section: “This manuscript uses three independent isogenic sets of TREM2 knockout iPSCs. These lines are each made on different patient iPSC backgrounds (two male, one female; all lines APOE3/3)”.

Also, in the Methods section, the authors explain that n refers to technical replicates for each experiment and that biological replicates are considered to be independent experiments performed at different dates, using - unless specified, but it is never specified - all three lines mentioned above. Since the observational unit (and thus the reported n) in these statistical analyses should be, for the experiments described here, the line and not the technical replicates, more details about how the technical and “biological” replicates as defined by the authors were

collapsed at the level of the observational unit, the line (e.g., by averaging measurements across technical replicates at different dates for each line, and then applying the statistical test for $n = 3$ where n is the number of lines, or using a hierarchical model?). More importantly, this detailed information about experimental design and statistical analysis should be extended to the *in vivo* experiments where a clear indication of how many lines were used is necessary. As an example, the heatmap in Fig. 1F shows 4 samples per combination of genetic background and stimulation: how many of these 4 samples are technical vs “biological” replicates? For each sample, the line used should be reported. In Fig. 1H, n is not reported.

We apologize for the prior lack of details regarding sample sizes for the various experiments. We have now updated the text to describe the sample sizes and clarify technical vs. biological replicates. Because the different patient backgrounds affect the baseline of many of our experiments (eg Line 2 has lower baseline phagocytosis of beta-amyloid than Line 1 in the WT cells alone), we did not collapse biological replicates to the line observational unit. Instead, we chose to show the data for one representative line. We would be happy to include data from separate lines as a supplemental figure if that would be helpful to the understanding of the paper.

For our *in vivo* experiments, we used $n=4$ WT animals and $n=6$ 5x animals. We now state this in the methods section. In all cases of *in vivo* work, the n refers to mouse as the biological unit. Additionally, as stated above, we have used two TREM2 knockout lines (GFP and RFP) for all *in vivo* experiments.

For Figure 1F (now Figure 1J), the replicates are technical replicates in one biological background. For Figure 1C a total of four iPSC lines were examined: two TREM2 KO and two isogenic TREM2 WT lines. Each pair of isogenic lines was generated from independent human subjects. For each line $n=3$ technical replicates were examined. For the functional experiments in Figure 2 and 3, we show representative results from 1 isogenic set; $n=4$ technical replicates. However, these experiments were repeated twice with each of the 2 additional lines on different days to confirm these functional phenotypes. For the caspase assay in Figure 2 and the dual-color APOE phagocytosis in Figure 3, we were only able to test 2 lines as the GFP endogenous fluorescence from Line 1 was incompatible with imaging those reporters. We state this now in the text.

4. Suppl. Table 1/2 format is very confusing. Please provide two separate sheets, one including information about all genes analyzed (not just the DEGs) along with gene ID (hugo, ensembl or entrez id), gene symbol, LFC and adjusted pvalue; and the other sheet including information about the significantly enriched genesets (GO, Kegg, etc), along with the pvalue of enrichment adjusted for the total number of terms/pathways tested (total number should be reported), total number of genes in the geneset as well as gene symbols and IDs of the DEGs that drives the enrichment. The adjusted pvalue threshold used for the genesets should be reported. The source and version/date of the genesets should also be reported. The authors should also refer to these analyses as geneset enrichment analyses and not GO enrichment analyses, since Kegg and other genesets were used in addition to GO genesets.

In response to this helpful comment, we have now included all the following information: in our supplemental tables; ID, gene symbol, logFC, adj p-value. We have also updated the tables shown for the previously collected data. Our data will also be available on GEO for any readers who wish to perform their own specific analyses. Geneset enrichment for GO Biological Process data is now included in the figure with all significant families displayed and thus no longer needed for the supplemental figure. We also state the cutoff values for fold change and significance (FDR) in the methods section.

We thank the reviewer for the reminder to include the source and version of datasets used and have now added this information to the manuscript.

5. For the experiments illustrated in Fig. 1, the authors used MAb-anti-TREM2 to stimulate TREM2 signaling as opposed to the TREM2 KO condition the authors used to obtain TREM2

loss-of-function. We argue that a better contrast to the TREM2 KO condition is over-expression of TREM2 in order to obtain gain-of-function. Indeed, the authors point out that MAb treatment may induce rapid internalization of TREM2 thus leading to a loss-of-function, particularly when considering the extremely transient activation of signaling downstream of TREM2 (minutes) compared to when the assays were performed (1 day post-treatment). Also, MAb-anti-TREM2 is not a disease-relevant stimulus (the authors used synaptosomes and fibrillar Abeta as disease-relevant stimuli in subsequent experiments). For these reasons, the transcriptome profiling would be a lot more informative if repeated using TREM2 over-expression and, for examples, synaptosomes, fibrillar Abeta or, even better, apoptotic cells as the disease-relevant stimulus, given that R47H mutation impairs binding of TREM2 to PtdSer and apoptotic cells. For the same reason, analysis of efferocytosis in Fig. 2 would be highly desirable. Also, how do the authors reconcile their findings that TREM2 loss-of-function (which is AD risk-increasing) leads to impaired synaptosome phagocytosis (which the authors claim to be a proxy for synaptic pruning) when increased synaptic pruning has been shown in mouse models to worsen disease-related phenotypes and proposed to be bad for AD, e.g., by Beth Stevens.

Thank you for these helpful comments. In response to this critique, we performed new sequencing experiments that including WT and TREM2 KO lines exposed to dead neurons. Unfortunately, we found that this stimuli does not induce a very TREM2-specific response. This likely results from the expression of many other microglial receptors that are known to respond to dead cell signals such as the TAM family of receptors. These new results have been included in Supplementary Figure 1 and Supplementary Table 2. However, given this finding we suggest that the TREM2 antibody provides a more specific stimuli to specifically examine TREM2 signaling without activating additional receptors. We have also added emphasis in the text that antibodies against TREM2 are currently being investigated by many pharma and biotech groups and indeed clinical trials concerning TREM2 antibody stimulation have begun. Thus, we conclude that antibody stimulation is definitely relevant to future therapeutic strategies and believe the audience for this journal will be able to use this sequencing dataset to further inform future studies of TREM2 antibody stimulation.

We have also now added the following additional discussion related to synaptosomes phagocytosis: "While decreased synaptic phagocytosis in TREM2 knockout microglia (modeled here with synaptosomes) may be surprising given that synaptic over-pruning is hypothesized to worsen AD progression, similar results have been shown in murine microglia^{39,60}. Additionally, it is possible that the ligand binding domain TREM2 mutations most strongly associated with AD, may not alter the recognition of this specific phagocytic substrate in the same way as a loss-of function deletion. Additionally, TREM2 function in synaptic pruning may be isolated to developmental stages *in vivo*. It has recently been shown in mice that TREM2 knockout results in hyper-connectivity in the brain and indeed lower expression of TREM2 is correlated with increased prevalence of autism spectrum disorder³⁹."

6. The authors keep referring to previous studies claimed to show decreased microglial migration in TREM2 KO compared to WT mice. However, these studies have only shown decreased number and altered morphology/activation state of microglial cells around plaques. This phenotype could be due to altered migration but also altered survival and/or proliferation, thus it should not be assumed, until directly observed or otherwise demonstrated, that this defect is the result of altered migration. Because the authors have also not demonstrated migratory deficits in their *in vivo* experiments, they should not use the words "migratory deficits/migration" when describing their results in Fig. 3A.

We confirm results from previous papers that TREM2 knockout microglia do have altered survival in Figure 2 and potentially with our single cell sequencing data (though we have moved discussion of the apoptotic cluster to supplemental figures and discussion). We have changed our language concerning the *in vivo* studies to highlight that the TREM2 microglia do not cluster around plaques and remain in homeostatic- looking morphology. We posit that the *in vitro*

experiments are indeed looking at migration towards the chemoattractants. We have shown that baseline motility of TREM2 microglia is not altered (via scratch wound assay, Figure 4D) thus would not be responsible for the increased numbers around migrating into the center of the microfluidic devices.

In Fig. 3A, the authors should specify what observational unit was used to derive the statistics shown in the bar plot (plaque, field, mouse? How many plaques/fields were used per mouse? Were the analyses blinded?). Related to this, and as previously requested, also the breakdown of the n into biological and technical and line replicates is needed for the reader to understand how many independent clonal cell lines were used in this experiment and whether the effect was consistent across the different clones. Also, when printed, the tiny colored spots are really difficult to see on the black background. Perhaps using false coloring may help the reader visualize the position of microglia in relation to the other features (plaques, circles, etc).

This is an excellent point. We have indeed confirmed results from previous studies that TREM2 knockout microglia exhibit altered survival in vitro (Figure 2) and thus reduced survival or proliferation could certainly contribute to the reduced association with plaques. We have therefore changed our language concerning the in vivo studies to no longer describe these findings as decreased migration. However, based on our in vitro studies of CXCR4-mediated chemoattraction we do posit that migration is impaired in TREM2 knockout microglia in vitro and thus may also be impaired in vivo.

In Fig. 3A, plaque morphology (and perhaps number) appears to be different between WT and TREM2 KO. Are these differences significant? Are these fields taken from matched regions in the brain? How do the authors explain such difference?

These fields are indeed taken from matched regions of the brain. However, since these experiments were done with co-transplants of WT and TREM2 KO cells together (expressing different fluorescent markers to tell them apart), the authors do not think it would be rigorous to comment on the plaque number and plaque morphology. Plaques are exposed to both WT and TREM2 knockout cells and thus their size cannot be compared between the two conditions in this study.

In Fig 3B, how do these concentrations and ratios of Abeta40 and 42 compared to what's measured in human AD brains? How do the authors explain such difference in response?

Here, our primary goal was to measure the iPS-microglia migration quantitatively in response to AD cues, particularly focusing on soluble abeta; however, we did not consider the migration effects regarding to ratios of Ab40 and Ab42, which are important for Ab plaque formation. Therefore, we picked up the concentrations of soluble Ab40 and Ab42 based on the lower limit of concentrations found in our in vitro AD models (Park et al.) and clinical studies and the upper limit of concentrations leading to migration. The stated concentration used refers to beta-amyloid plated in the central chamber, as these chambers are used to form a stable gradient, the concentration around microglia in the outer chamber is much lower. Additionally, these concentrations are similar to concentrations found in the CSF of patients with Alzheimer's disease (PMID:27504119, 28866757, and 25079805).

In Fig 3C, is the migration toward AD neurons vs healthy neurons due to apoptosis in the AD neurons cultured for 9w vs 3w? Since TREM2 has been shown to mediate the response of microglia to apoptotic cells, please repeat the experiment using live vs apoptotic cells (Jurkat cells for example) to ascertain whether the migratory response is due to Abeta or cues present on apoptotic cells in general. Also the title of Fig. 3 is misleading because microglia don't appear to respond robustly to healthy neurons and lack of TREM2 has not been tested in these experiments to ascertain whether it affects migration toward healthy neurons. This additional condition should be added to these experiments. Moreover, what are the levels of soluble Abeta

40 and 42 in the conditioned medium at the two different time points for the healthy and AD neurons?

In this regard, we agree that the cell death found in 9 week AD models can increase the migration significantly compared to 3 week AD models and healthy models. In a prior study that established and characterized this microfluidic model system, many factors were found to be engaged in the recruitment of microglia, not only soluble abeta but also other chemoattractants (e.g., ATP, CCL2, SDF1a) as well as dead cellular components (Park et al. Nature Neuroscience 2018; PMID 29950669). We observed the significant cell death in 9 week AD models compared to no cell death in 3 week AD models and 9 week healthy models which may play a role in this. The concentrations of beta-amyloid are described in Park et al. with 3-week old cultures reaching a concentration of 1.5 ng/mL abeta 40 and 0.4 ng/mL abeta 42. 9-week old cultures reached 2 ng/mL abeta 40 and 0.4 ng/mL abeta 42. However, we show in the subsequent figure that migration to the neural/glial cultures is likely due to CXCR4 SDF-1a engagement.

In Fig. 3D and E, if the authors want to make a more solid claim about the role of CXCR4 in the migratory deficit observed in Fig. 3C, they should deplete CXCL12 using an antibody and show the effect in the assays used in Fig. 3C.

This is an excellent point. To address this concern we have now quantified CXCR4 expression via two methods (ICC and flow cytometry). We also now have included migration experiments using a CXCR4 blocking agent (AMD3100). This additional experiment revealed that migration of WT cells towards AD neural cultures is indeed reliant on CXCR4 function and thus blocking CXCR4 is sufficient to block migration towards these cultures. In addition, we have provided more information about SDF1a expression in these models.

MINOR:

“Microglia perform key macrophage-like functions” -> Microglia are macrophages and perform the same core functions of all macrophages (e.g., efferocytosis and immune surveillance/response).

This sentence has been changed to: “As the primary immune cell of the brain, microglia perform key macrophage-functions such as phagocytosis of dead cell debris and protein aggregates, cytokine/chemokine signaling, and immune surveillance and response”.

“With the identification of several immune genes that alter risk for AD” -> AD GWAS studies identify loci, not genes. With a few exceptions, including TREM2, we don't have, at the moment, additional evidence to identify or prioritize causal genes in AD loci. Also, TREM2 is not microglia-specific and is expressed on several other tissue-resident macrophages, including non-parenchymal brain macrophages. A role for TREM2 function in this other myeloid cell populations in the modulation of AD risk cannot be excluded.

Thank you for this important point. This sentence has been changed to: “With the recent identification of several AD-risk loci near immune genes...”. We also agree that TREM2 is not exclusively expressed in microglia and have changed this sentence to “predominantly” expressed by microglia.

Main text refers to Suppl. Table 1 to include 444 DEGs yet the table contains 695 DEGs, this discrepancy should be explained and corrected. Main text refers to Suppl. Table 2 to include 114 DEGs yet the table contains 335 DEGs, this discrepancy should be explained and corrected.

Thank you for catching this. In this updated manuscript we have ensured that the text matches the supplemental tables. In addition, our supplemental tables now include expression of all genes, rather than just the differentially expressed genes so that readers may more easily work with the data and can make their own cutoff decisions if they so choose.

In Fig. 1F, why is gene expression reported as negative log of TPM+1 and why does this measure shows positive values? Normally expression in RNA-Seq experiments is reported as $\log(\text{TPM}+1)$ which is a positive number starting from zero for non-expressed genes.

Thank you for catching this! Our expression values in the heatmaps are median-centered for better visualization which is why positive and negative numbers become introduced. The figure legend now more accurately reports this.

In Fig. 1H one of the treatments is called 'no medium'. This is confusing. It should say -IL34,-MCSF,-TGFB1 or "no medium change". In an ideal experimental design, medium should have been changed with the same medium lacking the three growth factors to exclude medium changing as a factor in the experiment.

You are correct that the ideal experimental design is medium being changed but merely lacking the three growth factors. This is indeed what we did and we apologize that the figure legend does not reflect this. We have changed the legend to "-IL34; -MCSF; -TGFB1" and have clarified in the figure legend that basal medium was added in these conditions but there were no cytokines supplemented.

In Fig. 1, the graphical figure legends are confusing, with colors and gray scale trying to convey lighter and darker colors. The different colors should all be clearly explained or another graphical attribute (e.g., line pattern or shape) used to visualize WT vs TREM2 KO.

To address this issue, we have added a figure legend to the left plot, and changed the symbol shape for the TREM2 knockout lines. Furthermore, we have moved this data to its own figure in the hopes that this makes the data more accessible. For the bar graph, we have included additional labels of the WT and KO bars in order to help readers better understand this data.

In Fig. 2 the authors mention Zymosan as specific to Dectin-2 when it has been shown to be a ligand for Dectin-1 as well. Also both Dectins signal through Syk phosphorylation... how do the authors explain the discrepancy between TREM2 ligands and zymosan phagocytosis when inhibiting Syk phosphorylation?

The authors thank you for bringing up this point, and we have included text to discuss this in the paper. Interestingly, while you are correct that Dectin-1/2 stimulation does lead to SYK phosphorylation, it has been shown that the phagocytosis that results from stimulation of Dectins is not SYK dependent (Underhill et al. *Blood* 2005; Schorey et al. *Curr Drug Targets* 2008). By inhibiting SYK with R406 and not detecting any change in levels of Zymosan phagocytosis, we again prove that phagocytosis triggered from Dectin-1/2 is a SYK independent process.

Reviewers' Comments:

Reviewer #1:

Remarks to the Author:

McQuade and colleagues performed an elegant study addressing TREM2 function in the context of AD, using chimeric mice allowing analysis of hiPSC-derived human microglia in an amyloid mouse model for AD.

My comments were largely addressed satisfactorily, I support publication.

Reviewer #3:

Remarks to the Author:

The authors have been responsive to the prior critique. The revised manuscript is much improved in clarity and provides novel data regarding the potential role of CXCR4 in mediating deficits in migration of trem2^{-/-} microglia. Much of the remaining data while interesting is confirmatory of data obtained in mouse models with mouse microglia. I have no further comments.

Reviewer #1 (Remarks to the Author):

McQuade and colleagues performed an elegant study addressing TREM2 function in the context of AD, using chimeric mice allowing analysis of hiPSC-derived human microglia in an amyloid mouse model for AD.

My comments were largely addressed satisfactorily, I support publication.

Reviewer #3 (Remarks to the Author):

The authors have been responsive to the prior critique. The revised manuscript is much improved in clarity and provides novel data regarding the potential role of CXCR4 in mediating deficits in migration of trem2^{-/-} microglia. Much of the remaining data while interesting is confirmatory of data obtained in mouse models with mouse microglia. I have no further comments.

We kindly thank the reviewers both for their comments that have helped make this paper more suitable to publish in Nature Communications and thank them for increasing the rigor of our work.